# Can We Build a Monolithic Model for Fake Image Detection?
# SICA: Semantic-Induced Constrained Adaptation for Unified-Yet-Discriminative Artifact Feature Space Reconstruction

**Bo Du** [1 2]  **Xiaochen Ma** [3]  **Xuekang Zhu** [1 2]  **Zhe Yang** [1]  **Chaoqun Niu** [1]  **Jian Liu** [2]  **Ji-Zhe Zhou** [1]

## Abstract

Fake Image Detection (FID), aiming at unified detection across four image forensic subdomains, is critical in real-world forensic scenarios. Compared with ensemble approaches, monolithic FID models are theoretically more promising, but to date, consistently yield inferior performance in practice. In this work, we identify the intrinsic distinctness of artifacts across subdomains—a critical barrier we term the "Ji-Zhe phenomenon". Driven by this phenomenon, we diagnose the cause of this underperformance for the first time: the collapse of the artifact feature space. The core challenge for developing a practical monolithic FID model thus boils down to the "unified-yet-discriminative" reconstruction of the artifact feature space. To address this paradoxical challenge, we hypothesize that high-level semantics can serve as a structural prior for the reconstruction, and further propose Semantic-Induced Constrained Adaptation (SICA), the first monolithic FID paradigm. Extensive experiments on our *OpenMMSec* dataset demonstrate that SICA outperforms 15 state-of-the-art methods and reconstructs the target unified-yet-discriminative artifact feature space in a near-orthogonal manner, thus firmly validating our hypothesis. The code and dataset are available at: https://github.com/venus-guangjian/SICA_OpenMMSec.

## 1. Introduction

*"How wonderful that we have met with a paradox. Now we have some hope of making progress." - Niels Bohr*

---

[1]Sichuan University [2]Guangjian Team, Ant Group [3]The Hong Kong University of Science and Technology. Correspondence to: Ji-Zhe Zhou <jzzhou@scu.edu.cn>.

*Proceedings of the 43rd International Conference on Machine Learning*, Seoul, South Korea. PMLR 306, 2026. Copyright 2026 by the author(s).

*Table 1.* **Overview of dominant artifacts across different domains.** The distinct **Dependencies** indicate that artifact strategies tailored for one domain are often non-transferable to others, highlighting the heterogeneous nature of artifacts. △ indicates artifacts that are conceptually transferable but barely work in other subdomains, while ✗ denotes conceptually non-transferable artifacts.

| Domain | Dominant Artifacts | Dependency (Constraint) | Transfer? | Rep. Method |
|---|---|---|---|---|
| Deepfake | Blending Boundary | Face Swapping Mask | ✗ | Face X-ray (Li et al., 2020b) |
| | Frequency | Facial Region | △ | F3-Net (Qian et al., 2020) |
| | Physiology | Human Pulse (rPPG) | ✗ | FakeCatcher (Ciftci et al., 2020) |
| | Landmark | Facial Structure | ✗ | CSH (Hu et al., 2021) |
| | Semantic Motion | Lip-Voice Sync | ✗ | LipForensics (Haliassos et al., 2021) |
| AIGC | Global Spectrum | Checkerboard Pattern | ✗ | CNN-Gen (Wang et al., 2020) |
| | Fingerprint | GAN/Gen. Architecture | ✗ | Freq-Analysis (Frank et al., 2020) |
| | Reconstruction | Diffusion Prior | ✗ | DIRE (Wang et al., 2023) |
| IMDL | Noise Residual | Camera Sensor (PRNU) | △ | ManTra-Net (Wu et al., 2019) |
| | Edge | Boundary Inconsistency | ✗ | MVSS-Net (Chen et al., 2021) |
| Doc | Text Morphology | Font/Glyph Rendering | ✗ | DocTamper (Qu et al., 2023) |

Since its proposal, Fake Image Detection (**FID**) (Du et al., 2025) has garnered increasing attention, as it enables accurate detection in real-world forensic scenarios where faking methods are unknown *a priori*. In general, FID focuses on unified real-fake detection across image forensic subdomains, including **Deepfake** (Nguyen et al., 2019; Li et al., 2020b) (facial forgeries), **AIGC** (Ojha et al., 2023; Xi et al., 2023) (fully AI-generated images), **IMDL** (Zhou et al., 2023; Guillaro et al., 2023) (region-level manipulations in natural images) [1], and **Doc** (Qu et al., 2023; Chen et al., 2024) (document forgeries).

Despite its significance, the progress of building FID models remains sluggish. To date, the only widely-explored approach for FID is model ensembling (Huang et al., 2025). By leveraging specialized detectors from each subdomain, ensembling multiple off-the-shelf state-of-the-art (SoTA) models with a routing mechanism offers an intuitive solution. However, due to error propagation and the routing bottleneck, this ensemble strategy suffers from the "barrel effect" and fails to surpass or even match the performance of individual SoTA models within each subdomain.

In the meantime, a unified monolithic model is inherently free from the barrel effect, representing a more promising paradigm. **Then, can we build a monolithic model to**

---

[1]As most works do, we also categorize partial edits by generative models as IMDL in this paper.

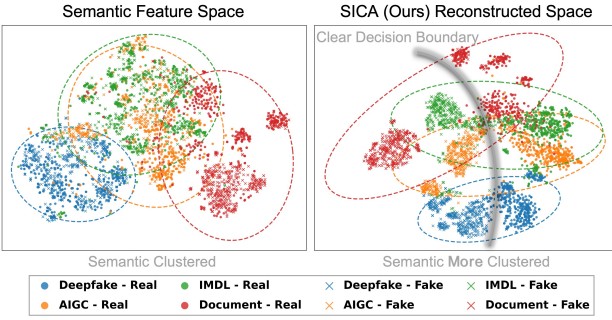

**Figure 1. t-SNE visualization of semantic feature space and SICA (ours) reconstructed space.** SICA leverages semantics to reconstruct unified-yet-discriminative artifact feature space.

### achieve accurate FID?

Directly answering this question by training monolithic models on subdomain-aggregated datasets consistently yields limited performance. Through extensive experiments and analysis, we discover that the real catch for this question is a previously unrecognized cross-domain behavior, which we term as the "Ji-Zhe phenomenon". As shown in Tab. 1, each subdomain exhibits its own dominant artifacts, which are highly domain-specific and typically non-transferable to other subdomains. For instance, the facial physiological artifacts, which rely on human pulse (rPPG), are conceptually impossible to be transferred or adopted in document images. Moreover, previous studies reveal that most methods tailored for specific subdomains perform worse than general backbones in FID (Du et al., 2025), such as ConvNeXt (Liu et al., 2022) and Swin Transformer (Liu et al., 2021b). These domain-specific and non-transferable characteristics of artifacts together constitute the **Ji-Zhe phenomenon: though termed the same, artifacts in FID are highly distinct across subdomains**.

Therefore, when adopting a generic backbone model in FID, this model always attempts to establish a unified representation for these heterogeneous artifacts. In other words, a monolithic model will project domain-specific, high-dimensional artifact features into a unified or domain-indiscriminative, low-dimensional representation space, leading to a **collapse of the feature space**. A direct evidence for this collapse is that involving extra training data from other subdomains will degrade model performance on the current subdomain (detailed in Sec. 5.3). As a result, a monolithic FID model can merely capture the shared or transferable components of these heterogeneous artifacts, discarding the primary and crucial domain-specific parts. Hence, even when training on large-scale datasets, a monolithic model consistently yields suboptimal results.

Based on the above observations, the real answer to building a monolithic FID model lies in **reconstructing the collapsed artifact feature space into a unified-yet-discriminative one**, in which a unified representation space aligns with the monolithic architecture, and discriminative artifact features address the collapse caused by the heterogeneous phenomenon. *While this "unified-yet-discriminative" requirement presents an apparent reconstruction paradox, a viable solution may exist.* Artifacts are heterogeneous across subdomains, but different subdomains also commonly contain different semantics (e.g., faces, documents, natural scenes). As illustrated on the left of Fig. 1, we conduct an empirical analysis on the semantic structure of FID data by defining semantic manifolds using CLIP (Radford et al., 2021) and visualizing them via t-SNE. The visualization confirms that semantic distributions are naturally clustered by subdomains: Deepfake and Doc are highly independent, while AIGC and IMDL exhibit partial overlap but remain largely distinct and discriminative. This discriminative nature is also observed in a more comprehensive FID dataset (*OpenMMSec*) we constructed later in this work.

Holding this subdomain-wise discriminative nature, **we hypothesize that high-level semantics**, which have long been regarded as inferior features in all subdomains of FID (Dong et al., 2023; Luo et al., 2021; Wang et al., 2020; Li et al., 2020a)**, can serve as the structural prior to reconstruct the unified-yet-discriminative artifact feature space.** Specifically, under this hypothesis, we speculate that a monolithic FID model can leverage semantic manifolds to anchor subdomains and further preserve domain-specific artifact features.

To validate our hypothesis, we seek to incorporate semantics explicitly into a monolithic FID model; however, this poses a technical dilemma: On one hand, naively incorporating semantic features as extra inputs (e.g., via fusion or concatenation) risks overfitting to semantic shortcuts (Zheng et al., 2024; Guillaro et al., 2025), which jeopardize model generalization. On the other hand, simply discarding semantic guidance leads to feature space collapse.

To tackle this dilemma, we propose to employ semantics not as direct input features, but as a guiding *inductive bias*[2] that structures the learning process itself. This leads to our **Semantic-Induced Constrained Adaptation (SICA)** paradigm. SICA is built upon two principles: (1) **semantic-induced**: using a frozen pretrained semantic backbone to provide a stable reference manifold; and (2) **constrained adaptation**: explicitly restricting the weight updates to be low-rank, since the low-rank update selectively bridges the semantic distribution gap between the reference manifold and training data, while preserving the reference manifold. This mechanism allows the model to establish an accurate

---

[2]Inductive bias refers to the set of assumptions that a learner uses to predict outputs for inputs not encountered during training (Battaglia et al., 2018).

inductive bias for artifact learning while mitigating semantic shortcuts (analyzed in Sec. 4.3). As shown on the right of Fig. 1, SICA effectively leverages the semantic manifold as the inductive bias and reconstructs the artifact feature space accordingly, enabling accurate real–fake classification.

To further validate the proposed hypothesis at a systematic level, we also construct *OpenMMSec* (detailed in Sec. 3) from existing public datasets, the first comprehensive and FID-tailored dataset. *OpenMMSec* aggregates data from **19** public forensic datasets and spans over **10** real-world datasets, containing over **330K** samples and covering all 4 subdomains with **98** image faking types.

Through extensive experiments on *OpenMMSec*, SICA: (1) captures artifacts as well as circumvents semantic shortcuts via low rank adaptation, successfully addressing the technical dilemma, as analyzed in Sec. 4.3; (2) stands as the first model that simultaneously outperforms 15 general backbones and subdomain detectors across all 4 evaluation metrics, as detailed in Sec. 5.2; and (3) reconstructs a unified-yet-discriminative artifact feature space in a near-orthogonal manner, as discussed in Sec. 5.3; thereby firmly validates our semantic structural prior hypothesis.

In short, we conduct a systematic exploration of artifact feature space, benefiting all four subdomains and other artifact-related forensic tasks. Our contributions are fivefold:

- **Identify the Unified-Yet-Discriminative Reconstruction Challenge:** We discover and formally define the Ji-Zhe phenomenon as the root cause of feature space collapse, framing the core challenge of monolithic FID as unified-yet-discriminative artifact feature space reconstruction.

- **Posit the Semantic Structural Hypothesis:** We propose that semantic manifolds provide the necessary structural prior to guide this reconstruction.

- **Introduce SICA - the First Monolithic Paradigm:** We establish SICA, which operationalizes this hypothesis via a frozen semantic backbone and constrained low-rank adaptation.

- **Construct a New *OpenMMSec* Benchmark:** We release *OpenMMSec*, the first large-scale comprehensive FID dataset (330K+ samples, 4 domains, 98 types) to enable systematic evaluation.

- **Deliver Rigorous Validation:** SICA outperforms 15 SoTA methods and successfully addresses the unified-yet-discriminative artifact feature space reconstruction challenge in a near-orthogonal manner, providing strong validation for our hypothesis.

## 2. Related Work

**Existing Works on Subdomains.** Image forensics has long been divided into four tasks: (1) Deepfake Detection (Sun et al., 2022; Nguyen et al., 2019), (2) AI-generated Image Detection (Wang et al., 2023; Cheng et al., 2025), (3) Image Manipulation Detection and Localization (Chen et al., 2021; Guillaro et al., 2023), and (4) Document Manipulation Localization (Qu et al., 2023; Chen et al., 2024). Although research within individual subdomains has progressed rapidly, each domain exhibits generalization limitations from a unified perspective, preventing direct application to FID. Details of these four domains can be found in Appendix C.

**Existing Works on FID.** FID has recently attracted increasing attention, focusing on subdomain-agnostic forgery detection (Du et al., 2025). Despite its importance, relevant research remains extremely scarce. Existing approaches rely on model ensembling, i.e., combining separate detectors trained on each domain (Huang et al., 2025). However, such an ensemble strategy inherently suffers from bottlenecks in both the detectors and the router. In contrast, a monolithic model is more promising.

**Existing Works on Artifacts.** The core of image forensics lies in exploiting manipulation traces, i.e., artifacts left by tampering operations. Each domain contains extensive research on domain-specific artifacts (Li et al., 2020b; Qian et al., 2020; Wang et al., 2023; Dong et al., 2022; Qu et al., 2023). However, they overlook the heterogeneous phenomenon of artifacts from a cross-domain perspective and thus provide limited guidance for FID.

## 3. The *OpenMMSec* Dataset

To conduct a systematic and fair study of FID, a comprehensive FID dataset with **diverse faking types**, **balanced data volume**, and **rich image sources** is first required. The previous FID benchmark (Du et al., 2025) concatenates limited public datasets from different subdomains (Rossler et al., 2019; Zhu et al., 2023; Wang et al., 2023; Dong et al., 2013; Qu et al., 2023) and thus suffers from the aforementioned three issues. To address this, we construct a comprehensive FID dataset, termed *OpenMMSec (Open Multi-Media Security)*, by sampling from existing public datasets across the four subdomains.

After a comprehensive survey of datasets across subdomains, *OpenMMSec*: (1) is organized by faking type, comprising **15** primary faking types and **98** fine-grained faking types (as shown in Fig. 2), covering comprehensive existing faking techniques across domains, **ensuring diverse faking types**; (2) carefully aligns the data volume across faking types to enable fair comparison, **ensuring balanced data volume**; and (3) integrates **19** subdomain datasets and spans over **10** real-world datasets, incorporating multiple subdomain

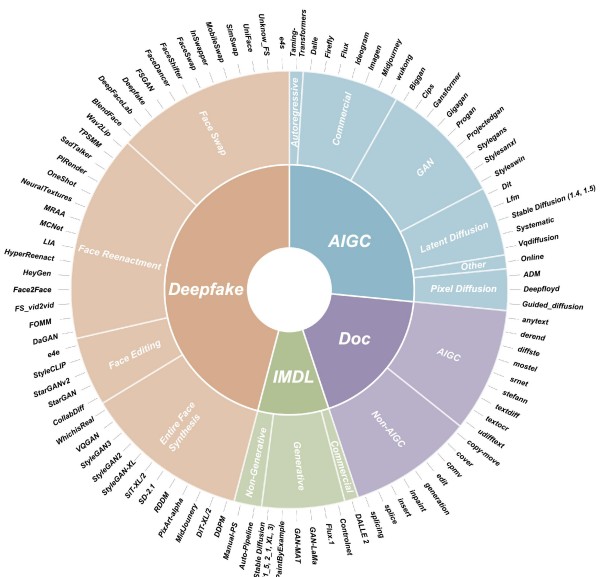

*Figure 2.* **Faking type overview of *OpenMMSec*.** Zoom in for better visualization of faking types.

sources for the same faking types, **ensuring rich image sources**. The datasets involved are listed in Appendix D.2.

Overall, *OpenMMSec* contains over 330K images and is organized into **15** primary faking types and **98** fine-grained faking types, with authentic images sourced from more than **10** real-world datasets. We present examples of several primary faking types from the four domains in Fig. 3. We retain pixel-level masks from the original datasets (IMDL and Doc) to support future research on localization. We carefully partition the data **by faking type** to fairly evaluate generalization of detectors to unseen fakings, with details provided in Sec. 5.1. Ethics statement and more construction details are provided in Appendix A and D respectively.

## 4. Methodology

In this section, we first formulate the artifact feature space collapse in FID. We then introduce the Semantic-Induced Constrained Adaptation (SICA) paradigm. Finally, we provide a spectral analysis of low-rank adaptation dynamics.

### 4.1. The Artifact Feature Space Collapse

FID aims to construct a unified classifier $\mathcal{F} : \mathcal{X} \rightarrow \{0, 1\}$ over an input space $\mathcal{X}$ composed of $K$ distinct subdomains $\mathcal{D} = \{D_k\}_{k=1}^{K}$ (e.g., Deepfake, AIGC). We define the *Ji-Zhe phenomenon* as the substantial distribution discrepancy between the intrinsic artifacts $\phi_{art}^{(k)}$ of these subdomains.

Optimizing a unified backbone $E_\theta$ forces these disjoint artifact distributions into a compact shared space $\mathcal{Z}$, causing **artifact feature space collapse**. Distinctive artifacts are compressed into overlapping regions:

$$\mathcal{Z}_{shared} \approx \bigcup_{k=1}^{K} E_\theta(\phi_{art}^{(k)}), \qquad (1)$$

leading to high inter-domain overlap. This interference dilutes decision boundaries, fundamentally limiting the model's ability to accommodate diverse artifacts.

### 4.2. The SICA Paradigm

To resolve the feature space collapse caused by artifact heterogeneity, we propose the **Semantic-Induced Constrained Adaptation (SICA)** paradigm.

**Semantic-Induced** employs a pretrained Vision Transformer (ViT) (Dosovitskiy et al., 2021) from CLIP (Radford et al., 2021) as the frozen backbone, denoted as $W_0$. We posit that $W_0$ provides a robust *semantic manifold* where inputs are naturally clustered by subdomain (e.g., faces, documents). This manifold serves as a stable coordinate system, allowing the model to anchor heterogeneous artifacts to their corresponding semantic contexts, thereby preventing the projection of high-dimensional features into a collapsed low-dimensional representation.

**Constrained Adaptation** utilizes low-rank adaptation (Hu et al., 2022) on self-attention layers, since the low-rank update selectively bridges the semantic distribution gap between the reference manifold and training data, while preserving the reference manifold. Specifically, we freeze $W_0$ and inject learnable updates $\Delta W$ parameterized by low-rank matrices: $h_{out} = W_0 h_{in} + \frac{\alpha}{r} B A h_{in}$. We illustrate SICA alongside two other weight adaptation methods, Fully Finetune (FFT) and Effort (Yan et al., 2024a), in Fig. 4. In contrast to FFT and Effort, SICA performs constrained alignment to co-adapt semantics and artifacts, thereby establishing an accurate inductive bias for artifact learning while mitigating semantic shortcuts.

### 4.3. Spectral Analysis of Low-rank Adaptation Dynamics

To empirically verify that low-rank mitigates semantic shortcuts to provide more space for artifact learning, we conduct a quantitative analysis of the learned weight updates using Singular Value Decomposition (SVD). Specifically, we analyze the **geometric relationship** between the pretrained weights $W_0$ and the learned update matrix $\Delta W$, comparing SICA with FFT and Effort (Yan et al., 2024a).

Formally, we denote the pretrained weights in CLIP as $W_0 \in \mathbb{R}^{m \times n}$. To extract the principal directions of $W_0$, i.e., the most important axes of transformation that capture the dominant semantic directions, we apply singular value decomposition (SVD) to decompose $W_0$:

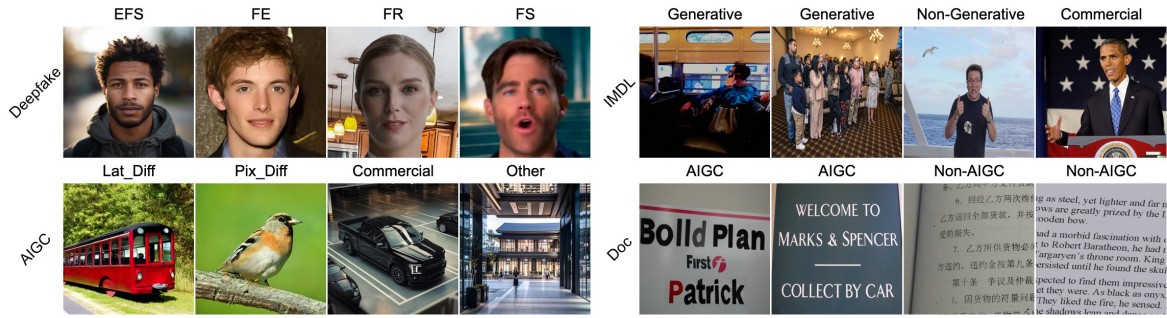

*Figure 3.* **Examples in *OpenMMSec*.**

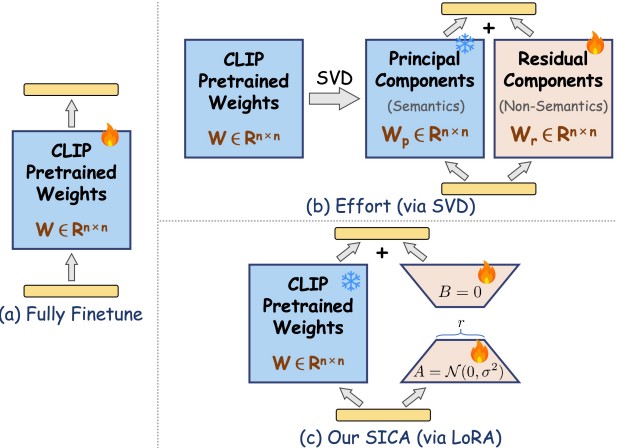

*Figure 4.* **Weight adaptation illustration of FFT, Effort, and the proposed SICA. (a) Fully Finetune (FFT)** updates the entire parameter space, risking semantic overfitting. **(b) Effort** (Yan et al., 2024a) explicitly decomposes weights via SVD into principal components (semantics) and residual components (artifacts), updating only the latter, yielding a rigid and suboptimal inductive bias. **(c) Our SICA** freezes the pre-trained weights and introduces a low-rank update to co-adapt semantics and artifacts.

$$W_0 = U\Sigma V^\top, \tag{2}$$

where $U \in \mathbb{R}^{m \times d}$ and $V \in \mathbb{R}^{n \times d}$ denote the orthogonal left and right singular matrices, respectively. $\Sigma \in \mathbb{R}^{d \times d}$ is the diagonal matrix of singular values. $d$ is the numerical rank (typically denoted as $r$ in SVD, but we use $d$ here to avoid confusion with the LoRA rank $r$).

We then select the top-$k$ subspace:

$$U_k = U_{[:,1:k]} \in \mathbb{R}^{m \times k}, \qquad V_k = V_{[:,1:k]} \in \mathbb{R}^{n \times k}. \tag{3}$$

We compute the update weights of the three schemes:

$$\begin{aligned}
\Delta W_{fft} &= W_{fft} - W_0, \\
\Delta W_{effort} &= U_{[:,k+1:]}\hat{\Sigma}V_{[:,k+1:]}^\top, \\
\Delta W_{sica} &= \frac{\alpha}{r}(BA),
\end{aligned} \tag{4}$$

where $\hat{\Sigma}$ is the learnable parameters, and $A \in \mathbb{R}^{r \times n}$, $B \in \mathbb{R}^{m \times r}$.

Given $U_k$, $V_k$, and the corresponding $\Delta W$ for the three, we can compute the projection operators of $\Delta W$ onto the left and right subspaces of $W_0$. We define the left/right subspace projection matrices as:

$$P_k^L = U_k U_k^\top, \qquad P_k^R = V_k V_k^\top. \tag{5}$$

Based on the projection matrices, we can compute the projection of the update matrix $\Delta W$ onto the two subspaces, corresponding to the principal subspace directions of $W_0$ (semantics), as well as the respective residuals (artifacts):

$$\begin{aligned}
\Pi_k^L(\Delta W) &= P_k^L \Delta W, \ R_k^L(\Delta W) = \Delta W - \Pi_k^L(\Delta W), \\
\Pi_k^R(\Delta W) &= \Delta W P_k^R, \ R_k^R(\Delta W) = \Delta W - \Pi_k^R(\Delta W).
\end{aligned} \tag{6}$$

Effort serves as a reference baseline, as its weight update $\Delta W_{effort}$ is theoretically orthogonal to the principal subspace of $W_0$. Consequently, we expect $\Delta W_{effort}$ to yield a near-zero projection onto the top-$k$ subspace.

With these matrices, we probe the relationships of $\Delta W_{sica}$ and $\Delta W_{fft}$ with respect to $W_0$ from two perspectives: *magnitude* and *direction*, using the **outside energy ratio** and **cosine similarity**, respectively.

**Outside energy ratio.** To quantify the proportion of update energy that lies outside the pretrained principal subspace, we compute the squared Frobenius norms of the residuals and of $\Delta W$:

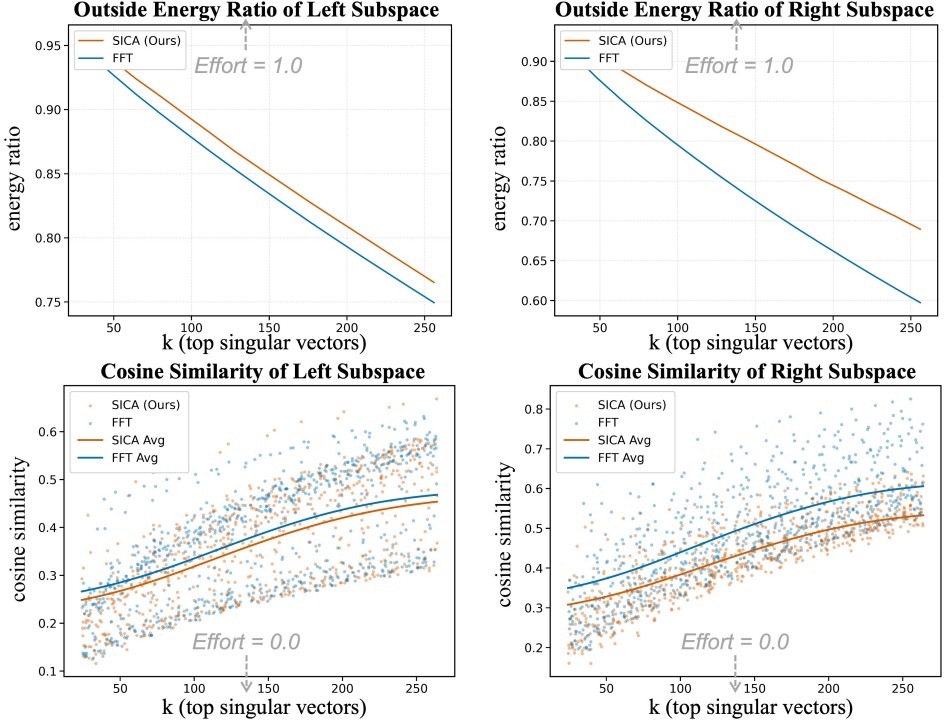

*Figure 5.* **Spectral analysis of left and right subspace via SVD.** SICA exhibits higher outside energy ratio and lower cosine similarity with respect to the dominant semantic subspace, proving to learn less semantics, thereby reducing the risk of semantic overfitting and enabling better artifact learning. Please refer to the main text for a more detailed description.

$$
\begin{aligned}
r_k^L &= \frac{\|R_k^L(\Delta W)\|_F^2}{\|\Delta W\|_F^2} = \frac{\|\Delta W - U_k U_k^\top \Delta W\|_F^2}{\|\Delta W\|_F^2}, \\
r_k^R &= \frac{\|R_k^R(\Delta W)\|_F^2}{\|\Delta W\|_F^2} = \frac{\|\Delta W - \Delta W V_k V_k^\top\|_F^2}{\|\Delta W\|_F^2}.
\end{aligned}
\tag{7}
$$

Finally, we average the results over multiple attention matrices $\bar{r}_k = \frac{1}{|\mathcal{M}|} \sum_{W \in \mathcal{M}} r_k(W)$.

We adopt weights trained on *OpenMMSec*. As shown on the top side of Fig. 5, Effort achieves an outside energy ratio consistently close to $1.0$. SICA exhibits a higher outside energy ratio in both the left and right subspaces, especially in the right subspace. This indicates that SICA tends to learn **outside the principal subspace** (i.e., artifacts) more strongly than FFT.

**Cosine Similarity.** While the outside energy ratio captures the magnitude of updates, we further investigate their directional relationships. For example, two update matrices with the same outside energy ratio may still have different dominant directions within the subspace. Therefore, we use cosine similarity to measure the consistency of update directions.

Accordingly, we vectorize the weight matrices and compute the similarity between the principal directions in the left and right subspaces and the original directions:

$$
\begin{aligned}
\mathrm{sim}_k^L &= cos(\mathrm{vec}(\Delta W), \mathrm{vec}(P_k^L \Delta W)) \\
&= \cos\big(\mathrm{vec}(\Delta W), \mathrm{vec}(U_k U_k^\top \Delta W)\big), \\
\mathrm{sim}_k^R &= cos(\mathrm{vec}(\Delta W), \mathrm{vec}(\Delta W P_k^R)) \\
&= \cos\big(\mathrm{vec}(\Delta W), \mathrm{vec}(\Delta W V_k V_k^\top)\big).
\end{aligned}
\tag{8}
$$

As shown on the bottom side of Fig. 5, Effort maintains a cosine similarity near $0$. Compared to FFT, SICA exhibits lower cosine similarity, especially in the right subspace, where SICA concentrates in a flatter low-similarity region. This indicates that SICA has a **weaker tendency to update semantic directions** during parameter adaptation.

From the above experiments, we conclude that compared to FFT, the weight updates $\Delta W$ of SICA via LoRA allocate a **larger outside energy** of the principal subspace spanned by the top-$k$ left/right singular vectors of $W_0$, while exhibiting **weaker directional alignment** with the corresponding subspace projections. Meanwhile, Effort strictly constrains updates to subspaces orthogonal to the dominant semantic directions. The spectral analysis proves that **SICA indeed reduces the risk of semantic overfitting and enables more effective artifact learning**.

*Table 2.* **Accuracy results on *OpenMMSec*.** The overall average is the macro-average of the domain averages. The best and second-best results are highlighted in **bold** and underlined, respectively.

| Method | Deepfake | | | | | AIGC | | | | | | | IMDL | | | Doc | | | Avg |
|---|---|---|---|---|---|---|---|---|---|---|---|---|---|---|---|---|---|---|---|
| | EFS | FE | FR | FS | Avg | GAN | Lat-Diff | Pix-Diff | AR | Comm | Other | Avg | Gen | Non-Gen | Avg | AIGC | Non-AIGC | Avg | |
| Resnet (He et al., 2016) | 71.3 | 65.4 | 70.0 | 64.7 | 67.8 | 77.4 | 65.2 | 75.7 | 78.5 | 72.7 | 75.5 | 74.2 | 80.3 | 74.2 | 77.3 | 61.9 | 81.5 | 71.7 | 72.7 |
| EfficientNet (Tan & Le, 2019) | 45.0 | 29.0 | 50.8 | 48.5 | 43.3 | 65.2 | 58.9 | 63.1 | 65.1 | 62.1 | 63.7 | 63.0 | 51.7 | 51.1 | 51.4 | 47.0 | 76.8 | 61.9 | 54.9 |
| CapsuleNet (Nguyen et al., 2019) | 63.4 | 58.7 | 66.5 | 59.6 | 62.0 | 79.2 | 63.1 | 77.5 | 81.3 | 73.2 | 77.8 | 75.4 | 79.2 | 75.3 | 77.2 | 59.8 | 84.8 | 72.3 | 71.7 |
| SegFormer (Xie et al., 2021) | 83.8 | 80.5 | 83.5 | 75.1 | 80.7 | 87.4 | 80.9 | 87.1 | 89.7 | 87.8 | 82.3 | 85.9 | 83.2 | 80.3 | 81.7 | 66.8 | 80.1 | 73.4 | 80.4 |
| Swin (Liu et al., 2021b) | 83.6 | 80.5 | 80.6 | 71.5 | 79.0 | 86.9 | 83.8 | 85.8 | 87.6 | 86.8 | 81.3 | 85.4 | 85.0 | 80.8 | 82.9 | 66.9 | 77.1 | 72.0 | 79.8 |
| ConvNeXt (Liu et al., 2022) | 80.5 | 84.0 | 83.9 | 72.4 | 80.2 | 89.0 | 79.8 | 87.9 | 92.0 | 89.6 | 83.4 | 87.0 | 83.6 | 80.2 | 81.9 | 65.0 | 75.3 | 70.1 | 79.8 |
| Recce (Cao et al., 2022) | 59.0 | 50.8 | 50.7 | 53.7 | 53.5 | 90.4 | 66.1 | 90.8 | 97.1 | 85.3 | 87.8 | 86.3 | 63.1 | 77.1 | 70.1 | 78.6 | 52.8 | 65.7 | 68.9 |
| Sia (Sun et al., 2022) | 64.0 | 63.5 | 73.1 | 63.7 | 66.1 | 81.6 | 68.7 | 80.2 | 84.1 | 79.2 | 80.4 | 79.0 | 71.9 | 63.8 | 67.9 | 64.4 | 83.4 | **73.9** | 71.7 |
| IML-ViT (Ma et al., 2023) | 66.6 | 64.9 | 74.6 | 66.0 | 68.0 | 83.3 | 70.6 | 81.8 | 86.1 | 81.5 | 80.9 | 80.7 | 81.5 | 78.9 | 80.2 | 68.9 | 78.5 | 73.7 | 75.7 |
| Trufor (Guillaro et al., 2023) | 73.7 | 72.7 | 77.1 | 65.4 | 72.2 | 83.7 | 77.9 | 83.0 | 86.0 | 84.0 | 80.3 | 82.5 | 82.7 | 78.3 | 80.5 | 58.1 | 86.6 | 72.3 | 76.9 |
| UnivFD (Ojha et al., 2023) | 57.0 | 44.8 | 57.6 | 57.4 | 54.2 | 84.1 | 68.3 | 81.7 | 84.9 | 76.9 | 82.9 | 79.8 | 65.7 | 76.2 | 70.9 | 39.0 | 86.3 | 62.7 | 66.9 |
| FFDN (Chen et al., 2024) | 73.7 | 75.5 | 81.2 | 69.5 | 75.0 | 93.6 | 89.2 | 93.9 | 96.1 | 94.5 | 86.8 | 92.4 | 84.7 | 80.6 | 82.6 | 64.8 | 75.3 | 70.0 | 80.0 |
| Effort (Yan et al., 2024a) | 87.1 | 87.2 | 80.7 | 85.0 | 85.0 | 86.6 | 71.3 | 84.6 | 86.3 | 78.8 | 83.9 | 81.9 | 81.7 | 85.7 | 83.7 | 61.1 | 78.1 | 69.6 | 80.1 |
| Mesorch (Zhu et al., 2025a) | 76.7 | 72.6 | 79.9 | 73.4 | 75.7 | 83.0 | 73.8 | 82.1 | 85.5 | 83.3 | 81.0 | 81.4 | 81.2 | 78.7 | 79.9 | 63.6 | 81.8 | 72.7 | 77.4 |
| CO-SPY (Cheng et al., 2025) | 75.1 | 71.3 | 77.0 | 65.5 | 72.2 | 84.7 | 80.8 | 84.2 | 86.2 | 85.7 | 78.5 | 83.3 | 77.8 | 74.2 | 76.0 | 61.9 | 78.0 | 70.0 | 75.4 |
| CLIP-FFT | 77.4 | 78.2 | 80.4 | 68.0 | 76.0 | 92.2 | 82.0 | 91.0 | 95.1 | 91.7 | 85.4 | 89.6 | 83.5 | 79.3 | 81.4 | 66.6 | 81.2 | **73.9** | 80.2 |
| **SICA (Ours)** | 91.3 | 89.4 | 86.5 | 86.6 | **88.4** | 96.3 | 94.5 | 94.7 | 96.7 | 94.8 | 86.9 | **94.0** | 85.4 | 85.1 | **85.3** | 69.2 | 78.4 | 73.8 | **85.4** |

# 5. Experiments

## 5.1. Setup

**Protocol.** Based on the 98 fine-grained faking types in *OpenMMSec*, to evaluate the generalization of detectors in FID, we carefully split 26 types for training (80K for training, 10K for validation) and the remaining 72 types for testing (240K). This protocol effectively evaluates the generalization to unseen faking types compared to the previous benchmark ForensicHub (Du et al., 2025) as discussed in Sec. 3. Nevertheless, we report the performance of SICA under ForensicHub in the Appendix E.3, and under subdomain benchmarks (Deepfake and AIGC) in Appendix E.4.

**Detectors.** We compare 15 methods, including visual backbones and SoTAs across four domains: *Backbones*: Resnet (He et al., 2016), EfficientNet (Tan & Le, 2019), SegFormer (Xie et al., 2021), Swin Transformer (Liu et al., 2021b), ConvNeXt (Liu et al., 2022); *Deepfake*: CapsuleNet (Nguyen et al., 2019), Recce (Cao et al., 2022), Sia (Sun et al., 2022); *IMDL*: IML-ViT (Ma et al., 2023), Trufor (Guillaro et al., 2023), Mesorch (Zhu et al., 2025a); *AIGC*: UnivFD (Ojha et al., 2023), Effort (Yan et al., 2024a), CO-SPY (Cheng et al., 2025); *Doc*: FFDN (Chen et al., 2024). Details of detectors can be found in Appendix D.4. We use ForensicHub (Du et al., 2025) as the codebase for implementation.

**Implementation Details.** We adopt CLIP ViT-L/14 (Radford et al., 2021) as the backbone, with LoRA (Hu et al., 2022) parameters set to $r = 8$ and $\alpha = 16$. More details are provided in Appendix E.1.

## 5.2. Generalization Comparison

We report results for each domain at the level of primary faking types. We report Accuracy (ACC) in Tab. 2, with AUC, AP, and F1 reported in the Appendix Tab. 6, Tab. 7, and Tab. 8 respectively. We compute the domain-wise averages

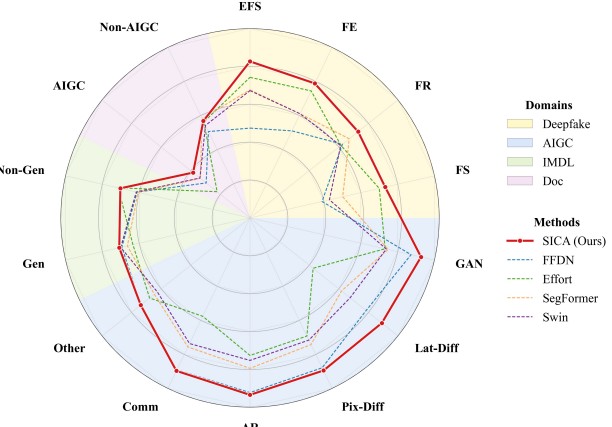

*Figure 6.* **Generalization comparison on diverse faking types.** SICA consistently outperforms top-performing detectors across 14 test faking types.

(shown in the gray regions of the table) and further calculate the macro-average across domains as the overall performance. The results show that the proposed SICA achieves three best and one second-best performances across the four domains, and obtains the best overall average. Moreover, SICA consistently outperforms others across all primary faking types, as shown in Fig. 6, demonstrating its superior generalization.

## 5.3. Feature Space Reconstruction Analysis

We further investigate the feature space reconstruction of SICA by comparing FFT. Specifically, we partition the training data by domain and adopt two training settings: *single-domain* (SD), where the model is trained using data from only one domain, and *leave-one-domain-out* (LODO), where the model is trained using data from all domains except the held-out one. We then evaluate the models on all out-of-distribution data from each of the four domains and

*Table 3.* Cross-domain performance comparison under different training strategies. Rows indicate training domains and columns indicate test domains. The last row reports unified training results.

| Train | Deepfake | AIGC | IMDL | Doc |
|---|---|---|---|---|
| Deepfake | **0.8183** | 0.7639 | 0.4605 | 0.3488 |
| AIGC | 0.5704 | **0.9291** | 0.4161 | 0.5263 |
| IMDL | 0.5862 | 0.4706 | **0.8535** | 0.7870 |
| Doc | 0.5604 | 0.4474 | 0.4843 | **0.8657** |
| Unified | 0.7900 | 0.8987 | 0.8223 | 0.8080 |

(a) FFT-SD

| Train | Deepfake | AIGC | IMDL | Doc |
|---|---|---|---|---|
| Deepfake | **0.9235** | 0.6497 | 0.5612 | 0.4455 |
| AIGC | 0.5496 | **0.9623** | 0.4205 | 0.6425 |
| IMDL | 0.8093 | 0.5869 | **0.8817** | 0.6980 |
| Doc | 0.4773 | 0.5032 | 0.5467 | **0.8893** |
| Unified | 0.9234 | 0.9572 | 0.8804 | 0.8646 |

(b) SICA-SD

| Train Wo. | Deepfake | AIGC | IMDL | Doc |
|---|---|---|---|---|
| Deepfake | **0.7660** | 0.9642 | 0.8662 | 0.8807 |
| AIGC | 0.9191 | **0.6491** | 0.8730 | 0.8622 |
| IMDL | 0.9096 | 0.9541 | **0.4674** | 0.8575 |
| Doc | 0.9221 | 0.9532 | 0.8812 | **0.7081** |
| Unified | 0.9234 | 0.9572 | 0.8804 | 0.8646 |

(c) SICA-LODO

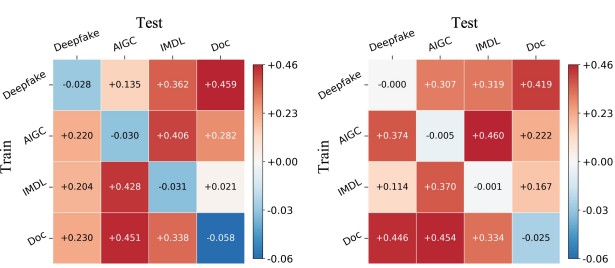

*Figure 7.* **Heatmaps of the performance differences between single-domain and unified training for FFT and SICA.**

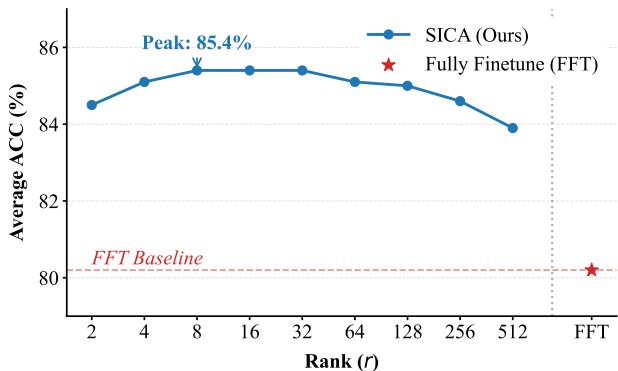

*Figure 8.* **Ablation study of rank.**

report AUC scores. The results are shown in Tab. 3.

The results indicate that, in most cases, artifacts do not transfer across domains, where training on one domain provides little benefit or conflict to another (Tab. 3 (a), (b)) (AUC of 0.5 corresponds to random prediction). Even when the training data are expanded to the LODO setting (Tab. 3 (c)), no substantial improvement is observed, suggesting that artifacts cannot be shared across domains. This further confirms the **heterogeneity of artifacts across domains**.

We compare the performance of models trained on the unified data, as shown in the last row of Tab. 3. For more intuitive visualization, we plot heatmaps of the performance differences between single-domain and unified training for the two training strategies, as shown in Fig. 7.

Focusing on the diagonal entries to compare FFT with SICA, FFT exhibits a substantially larger performance drop, whereas SICA shows almost no performance degradation. This indicates that SICA maps artifacts from different domains into near orthogonal subspaces, verifying that **SICA effectively reconstructs the artifact feature space**.

### 5.4. Ablation Study

**Ablation Study of Rank.** In Sec. 4.3, we analyze SICA via spectral analysis and show that low-rank achieves less semantic learning. To further investigate this behavior, we conduct a comprehensive ablation study on the LoRA rank. The results in Fig. 8 show that ACC first increases and then decreases as the rank grows, forming a peak plateau when the rank ranges from 8 to 32. This indicates that a

small rank limits artifact modeling, while an excessively large rank weakens the constraints and increases the risk of semantic overfitting, ultimately degrading performance.

*Table 4.* **Ablation of pretrain backbones.** The evaluation metric used is accuracy (ACC).

| Backbone | Pretrained | Deepfake | AIGC | IMDL | Doc | Avg |
|---|---|---|---|---|---|---|
| Resnet-50 | ImageNet-1K | 67.8 | 74.2 | 77.3 | 71.7 | 72.7 |
| Resnet-50 | CLIP | 73.6 | 88.1 | 79.7 | 71.8 | 78.3 |
| ViT-L/14 | DINOv2 | 86.8 | 86.4 | 83.8 | 74.7 | 82.9 |
| ViT-L/16 | SigLIP | 81.7 | 88.8 | 83.9 | 74.4 | 82.2 |
| ViT-L/14 | CLIP | 88.4 | 94.0 | 85.3 | 73.8 | 85.4 |

**Ablation Study of Pretrained Backbones.** We present results with different pretrained backbones in Tab 4. From the ResNet (He et al., 2016) results, using CLIP-pretrained weights (Radford et al., 2021) significantly improves performance compared to ImageNet-1K (Deng et al., 2009) (from 72.7 to 78.3), effectively validating the importance of strong semantic manifold. Moreover, ViT with CLIP pretraining outperforms DINOv2 (Oquab et al., 2023) and SigLIP (Zhai et al., 2023), indicating that the global semantics learned through image–text alignment in CLIP serve as a more advantageous structural prior for SICA.

### 5.5. Failure Case

We present several failure cases in Fig. 9. SICA fails in scenarios involving complex semantic contexts or those poorly covered by the pre-training data, as the semantic

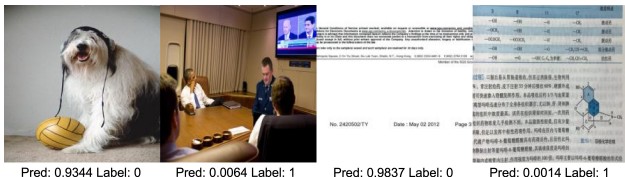

*Figure 9.* **Failure case of SICA.**

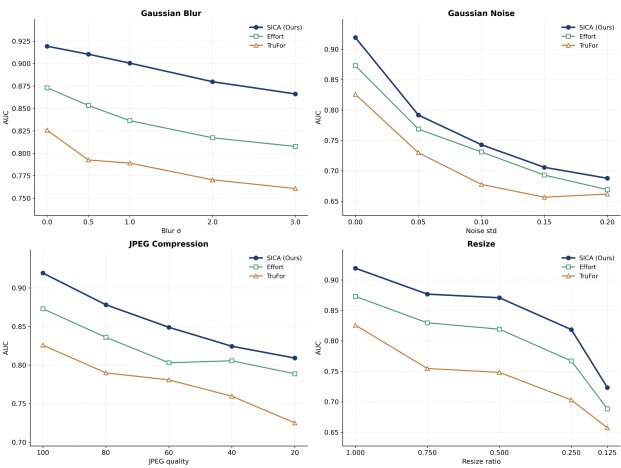

*Figure 10.* **Robustness experiment.**

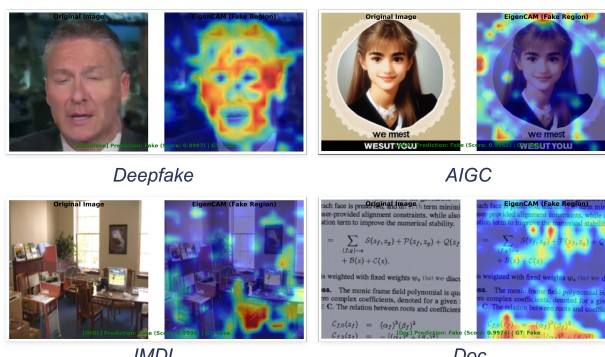

*Figure 11.* **Attention map of fake images across four domains.**

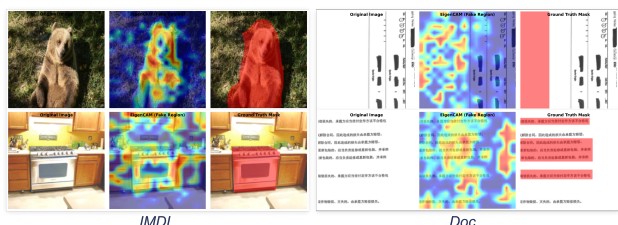

*Figure 12.* **Attention map of fake images in IMDL and Doc with corresponding regions.**

reference manifold becomes unreliable. This limitation could be addressed by stronger semantic backbones in the future.

### 5.6. Robustness Experiment

We randomly sampled 2,500 real and 2,500 fake images in OpenMMSec to evaluate SICA's robustness against two SOTAs (Effort and Trufor) across four post-processing methods: Gaussian Blur, Gaussian Noise, JPEG Compression, and Resize. The results provided in Fig.10 demonstrate SICA's superior robustness under varying post-processing conditions.

### 5.7. Attention Map Visualization

To provide visual interpretability, we employed Eigen-CAM (Gildenblat & contributors, 2021) to extract SICA's attention maps for fake images across all four subdomains (Fig.11 and Fig.12). Although SICA is an image-level detection model rather than a pixel-level localizer—naturally yielding less precise map-to-mask alignments—the visualizations confirm it accurately identifies subdomain-specific artifacts consistent with prior studies. Specifically, attention is concentrated on the facial region for Deepfake, distributed globally for AIGC, and localized on the tampered regions for IMDL and Doc. This visual evidence directly corroborates our core claim: SICA successfully reconstructs the artifact feature space by accommodating domain-specific

heterogeneous artifacts without cross-domain interference.

## 6. Conclusion

In conclusion, this paper identifies artifact feature space collapse as the primary obstacle to achieving high-performance monolithic Fake Image Detection. By introducing Semantic-Induced Constrained Adaptation (SICA), we effectively reconstruct a unified-yet-discriminative artifact feature space. Comprehensive evaluations on our newly constructed Open-MMSec benchmark—the first large-scale FID dataset comprising over 330K samples across 98 fine-grained faking types—demonstrate that SICA outperforms current state-of-the-art methods and validates the efficacy of leveraging high-level semantics as structural priors. Our work provides a promising and scalable paradigm for real-world universal image forensics.

## Acknowledgment

This work is supported by the National Natural Science Foundation of China (NSFC), Youth Fund (Category C)(No.62506251) and Sichuan Province Major Special Project (2024ZDZX0001-3). Furthermore, this work was conducted during an internship at Guangjian Team, Ant Group, and the authors would like to thank Ant Group for providing the computational resources and support. We also thank Chengdu Haiguang Integrated Circuit Design Co., Ltd. for providing the HYGON K100AI DCU computations.

## Impact Statement

This work contributes to the advancement of image forensics, specifically addressing the critical challenge of detecting sophisticated fake images across diverse domains. By providing a unified and generalizable detection framework, our research aims to mitigate the spread of misinformation, protect intellectual property, and safeguard the integrity of visual media in society. We acknowledge the potential dual-use risk where our detection methods could be exploited by adversaries to enhance the realism of manipulated images via adversarial training. To mitigate potential misuse, we will release our code under a strict license that limits usage to academic and non-commercial research purposes.

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

# A. Data Availability and Ethics Statement

**Source Compliance:** *OpenMMSec* is a benchmark suite constructed from publicly available forensic datasets and academic repositories. We strictly adhere to the original licenses and redistribution policies of all source materials. For data sources that prohibit redistribution, we do not re-host the raw samples; instead, we provide comprehensive metadata (e.g., URLs, IDs) along with automated construction scripts. This ensures that researchers can independently obtain the original data from the primary providers and faithfully reproduce our benchmark splits and preprocessing steps.

**License Terms:** *OpenMMSec* is released under the **Creative Commons Attribution-NonCommercial 4.0 International (CC BY-NC 4.0)** license. Details of the license can be seen in `https://creativecommons.org/licenses/by-nc/4.0/`. All third-party data remain subject to their original licenses and terms, including non-commercial and/or non-redistribution constraints.

**Usage Restrictions:**

- **Academic Use Only:** Access to the dataset is granted exclusively to researchers from educational institutes and non-profit organizations. Commercial use of any part of this dataset is strictly prohibited.

- **Redistribution:** Users are not permitted to re-host or redistribute the dataset without explicit permission.

- **Opt-out Policy:** We respect the intellectual property and privacy rights of all original data owners. Should any copyright holder or subject wish to have their data removed from *OpenMMSec*, they may contact the authors, and we will promptly remove the relevant samples from our distribution.

**Availability:** The full dataset and metadata (e.g., URLs/IDs), along with the data construction scripts and splitting protocols, will be publicly released upon the acceptance of this paper.

# B. Supplementary of Semantics as the Structure Prior

Directly projecting heterogeneous artifacts from different domains into a shared subspace leads to feature space collapse. However, we observe that semantics across domains can serve as an effective structural prior. Specifically, in subdomain settings, method designs are already implicitly based on domain-specific semantic priors. For example, Deepfake detection focuses exclusively on facial images and thus leverages facial semantics as priors. Therefore, explicitly incorporating semantics as a structural prior in FID can help reconstruct the artifact feature space, as shown in Fig. 13.

Intuitively, semantics differ substantially across domains. For Deepfake and Doc, the tasks are restricted to faces and text images, respectively, resulting in highly specific and compact semantic distributions. AIGC and IMDL both target general scenes. However, AIGC involves full-image generation and thus exhibits a broader semantic distribution, whereas IMDL performs partial manipulation on real images, leading to a more compact distribution that remains closer to that of real images.

# C. Detail of Subdomains

Although image forensics fundamentally aims to distinguish real images from manipulated ones, it has long been divided into four subdomains due to differences in targets and manipulation techniques. To the best of our knowledge, research on unified FIDL is sparse, with existing methods relying either on multi-model ensembles (Huang et al., 2025) or proprietary, closed-source detection models (Team, 2026).

### C.1. Deepfake

Deepfake detection primarily focuses on whether faces in images or videos have been manipulated, and existing methods can be broadly categorized into naive, spatial, and frequency-based detectors (Yan et al., 2023). Deepfake detection designs artifact extraction strategies with facial priors, such as biometric cues (Ciftci et al., 2020), landmarks (Hu et al., 2021), and facial motion patterns (Haliassos et al., 2021).

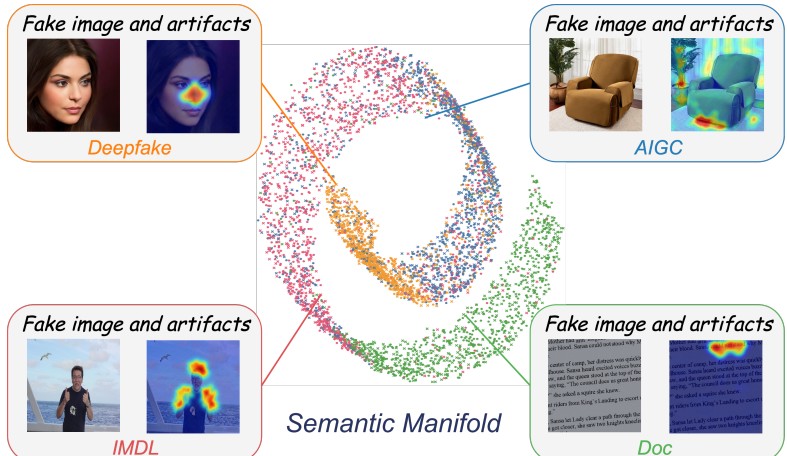

*Figure 13.* **Illustration of leveraging semantics as the structured prior to reconstruct artifact feature space.**

## C.2. AIGC

AI-generated image detection targets images that are entirely synthesized by generative models. As generative models have rapidly advanced in recent years, they can now produce fake images that are nearly indistinguishable to the human eye. AIGC detection primarily relies on model-specific artifacts, such as reconstruction anomalies in diffusion models (Wang et al., 2023) and frequency irregularities in GANs (Wang et al., 2020), and has attracted substantial attention in recent years.

## C.3. IMDL

Image manipulation detection and localization focuses on partial edits in natural images (Zhu et al., 2025b) and thus produces two outputs: an image-level prediction of real or fake, and a pixel-level classification of manipulated regions (Ma et al., 2024). IMDL typically relies on hand-crafted artifact extraction strategies, such as noise (Wu et al., 2019; Zhou et al., 2018), frequency (Kwon et al., 2022; Wang et al., 2022a), and edge artifacts (Chen et al., 2021; Ma et al., 2023). In recent years, the rise of generative models has posed new challenges for IMDL, as partial-region manipulations based on generative models have also spurred the development of new detection methods (Wang et al., 2025).

## C.4. Doc

Document image manipulation localization targets text-level manipulations. Such manipulations often manifest as inconsistencies between text and background (Shao et al., 2023); however, compared to other domains, the artifacts are much more subtle, as document images contain far less information than complex faces or natural scenes. Recent methods typically exploit block artifact grids (Qu et al., 2023) or inconsistencies among characters (Wang et al., 2022b).

However, unified FID without distinguishing domains is becoming increasingly important (Du et al., 2025; Huang et al., 2025). On the one hand, domain-specific approaches create silos that hinder scientific progress; on the other hand, the rapid evolution of generative models is blurring the boundaries between domains. In real-world scenarios, forensic detectors cannot know the domain of a test image in advance, further highlighting the necessity of FID.

## D. Construction Detail of *OpenMMSec*

### D.1. Detail of Construction Basis

Currently, there is no public dataset covering all four domains, and each domain has constructed its own domain-specific datasets. To fairly and comprehensively validate the training paradigm proposed in this paper and to facilitate future unified detection research, we aim to construct a unified dataset with the following properties: (1) complete coverage of all four domains with as many faking types as possible; (2) easy extensibility to accommodate new faking methods; and (3) coverage of as many real-image sources as possible to mitigate content bias.

To this end, we construct *OpenMMSec* by sampling over 330K images from 19 public datasets across domains, with **faking**

| Data Partition | #Source Datasets | Primary Type | Fine-grained Type | Total | Real | Fake |
|---|---|---|---|---|---|---|
| Deepfake | 6 | 4 | 45 | 94636 | 29000 | 65636 |
| AIGC | 3 | 6 | 26 | 91048 | 46000 | 46048 |
| IMDL | 7 | 3 | 9 | 98914 | 47914 | 51000 |
| Doc | 3 | 2 | 18 | 48985 | 6388 | 42597 |
| Total | 19 | 15 | 98 | 333583 | 129302 | 204281 |
| Train | / | 12 | 26 | 81632 | 34736 | 46896 |
| Validation | / | / | / | 8240 | 3864 | 4376 |
| Test | / | 14 | 72 | 243711 | 90702 | 153009 |

*Table 5.* **Statistics of *OpenMMSec*.** We split the train and generalization test sets according to fine-grained types.

**type** as the primary dimension. We organize faking methods into 15 primary faking types and 98 fine-grained faking types, and carefully control the number of samples per type to ensure balanced coverage. For faking types that appear in multiple source datasets, we sample from each source to increase diversity. During sampling, we strictly use the fake images and the corresponding source real images to avoid dataset bias. For IMDL and Doc sources, we also retain pixel-level mask annotations to support future localization tasks. Based on the faking-type taxonomy, we select 26 fine-grained faking types for training, split into training and validation sets with a 9:1 ratio. The remaining 72 fine-grained faking types are used as the test set to evaluate generalization. The detailed statistics of OpenMMSec are presented in Tab. 5.

In addition, we place particular emphasis on the sources of real images. Previous studies (Zheng et al., 2024) have shown that limited real-image sources can introduce content bias and hinder the learning of artifacts. Accordingly, our sampled datasets cover more than ten real-image datasets, including multiple online sources, ensuring high diversity and comprehensive coverage. We illustrate the real-image sources covered in *OpenMMSec* in Fig. 14.

### D.2. Summary of Datasets Involved in *OpenMMSec*

The public forensic datasets involved in constructing *OpenMMSec* are: (1) *Deepfake*: FaceForensics++ (Rossler et al., 2019), FaceShifter (Li et al., 2019), DFD (Google AI Blog, 2019), DFDC (Dolhansky et al., 2020), CelebDF-v2 (Li et al., 2020c), DF40 (Yan et al., 2024b); (2) *AIGC*: DiffusionForensics (Wang et al., 2023), GenImage (Zhu et al., 2023), Community Forensics (Park & Owens, 2025); (3) *IMDL*: CASIAv2 (Dong et al., 2013), IMD2020 (Novozamsky et al., 2020), tamperCOCO (Kwon et al., 2022), MIML (Qu et al., 2024), Autosplice (Jia et al., 2023), GRE (Sun et al., 2024), OpenSDI (Wang et al., 2025); (4) *Doc*: Doctamper (Qu et al., 2023), OSTF (Qu et al., 2025), RTM (Luo et al., 2025).

Beyond the diverse faking types, *OpenMMSec* also covers a comprehensive set of real image sources, including ImageNet (Deng et al., 2009), COCO (Lin et al., 2014), UADFV (Li & Lyu, 2018), CelebA (Liu et al., 2018), FaceForensics++ (Rossler et al., 2019), DFD (Google AI Blog, 2019), FFHQ (Karras et al., 2019), CelebDF (Li et al., 2020c), Visual News (Liu et al., 2021a), VFHQ (Xie et al., 2022), LAION (Schuhmann et al., 2022), Megalith-10M (madebyollin, 2024), as well as images collected from online sources.

### D.3. Detail of Domain Datasets for *OpenMMSec*

#### D.3.1. DEEPFAKE

**FaceForensics++** (Rossler et al., 2019) is a standardized benchmark for facial forgery forensics, constructed from approximately 1,000 real-world videos and augmented with forged samples generated by multiple automated face manipulation methods. The authors further provide different compression levels and settings to systematically evaluate the robustness of detection methods under compression and resolution variations.

**FaceShifter** (Li et al., 2019) is commonly used as a representative face-swap forgery pipeline to enrich the diversity of manipulation types.

**DFD** (Google AI Blog, 2019) released by Google is part of a data contribution initiative for deepfake detection, aiming to provide the research community with deepfake samples for training and evaluation, and has been integrated into the FaceForensics benchmark ecosystem to facilitate reproducible evaluation.

**DFDC** (Dolhansky et al., 2020) is a large-scale face-swap video dataset constructed for the Kaggle Deepfake Detection Challenge, reported to contain over 100K video clips from thousands of participants and generated using multiple forgery

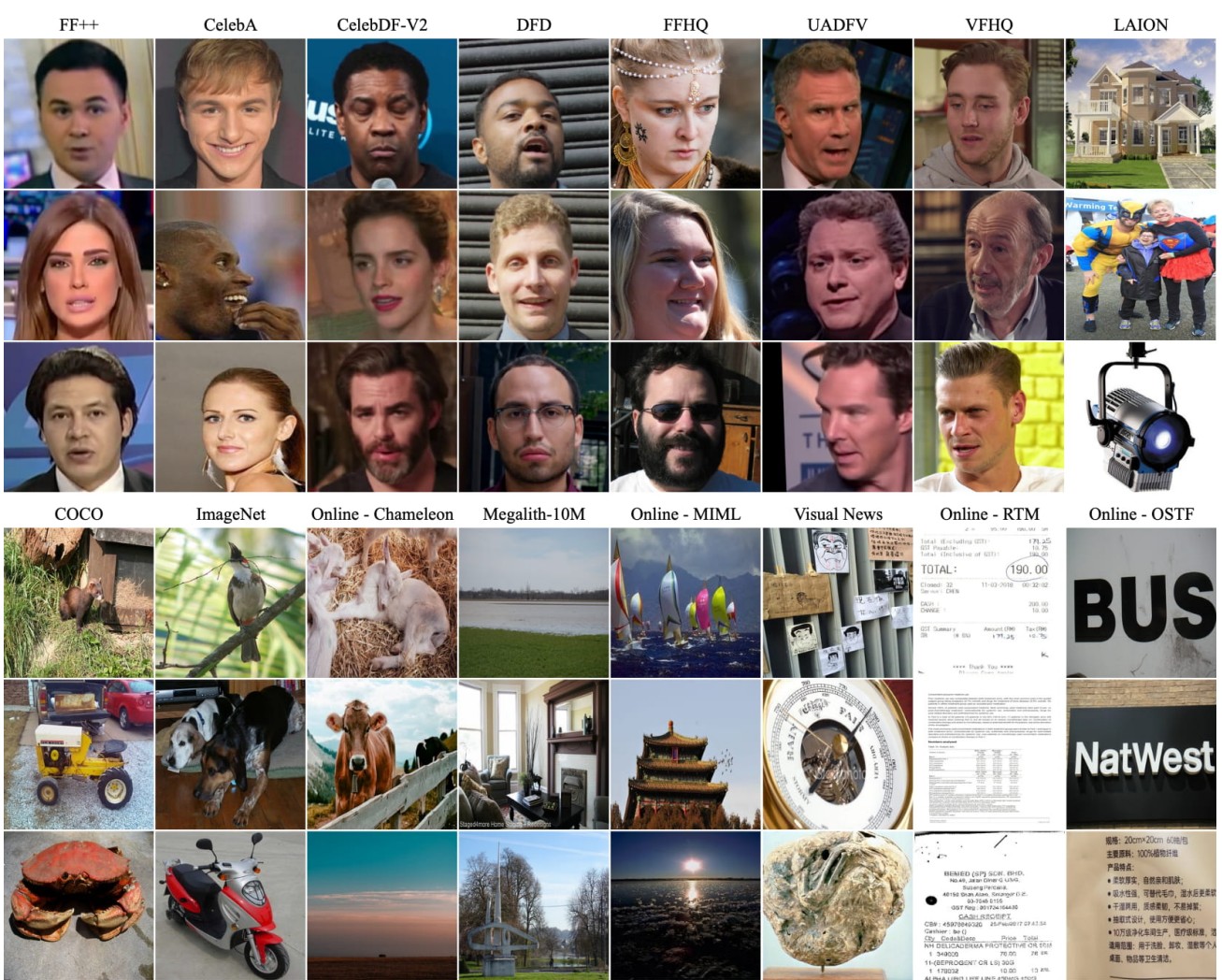

*Figure 14.* **An overview of the real-image datasets included in *OpenMMSec*.**

*Table 6.* **AUC results on *OpenMMSec*.** The overall average is the macro-average of the domain averages. The best and second-best results are highlighted in **bold** and underlined, respectively.

| Method | Deepfake | | | | | AIGC | | | | | | | IMDL | | | Doc | | | Avg |
|---|---|---|---|---|---|---|---|---|---|---|---|---|---|---|---|---|---|---|---|
| | EFS | FE | FR | FS | Avg | GAN | Lat-Diff | Pix-Diff | AR | Comm | Other | Avg | Gen | Non-Gen | Avg | AIGC | Non-AIGC | Avg | |
| Resnet (He et al., 2016) | 78.0 | 58.6 | 75.3 | 68.4 | 70.1 | 86.0 | 70.6 | 71.0 | 97.7 | 70.8 | 79.3 | 79.2 | 88.9 | 75.4 | 82.2 | 53.6 | 86.5 | 70.1 | 75.4 |
| EfficientNet (Tan & Le, 2019) | 58.7 | 52.8 | 59.2 | 59.2 | 57.5 | 73.7 | 63.2 | 63.1 | 84.3 | 61.8 | 67.7 | 69.0 | 52.5 | 50.6 | 51.2 | 46.7 | 72.5 | 59.6 | 59.4 |
| CapsuleNet (Nguyen et al., 2019) | 67.7 | 52.4 | 72.9 | 62.2 | 63.8 | 82.2 | 65.3 | 65.4 | 90.1 | 61.9 | 78.8 | 74.0 | 85.7 | 71.3 | 78.5 | 39.6 | 86.1 | 62.9 | 69.8 |
| SegFormer (Xie et al., 2021) | 89.5 | 80.5 | 89.8 | 81.9 | 85.4 | 91.8 | 88.0 | 84.6 | 99.8 | 91.8 | 77.1 | 88.9 | 89.7 | 78.5 | 84.1 | 59.5 | 87.9 | 73.7 | 83.0 |
| Swin (Liu et al., 2021b) | 91.6 | 85.2 | 86.2 | 77.3 | 85.1 | 93.6 | 91.5 | 85.7 | 99.9 | 93.6 | 80.1 | 90.7 | 92.3 | 80.7 | 86.5 | 61.4 | 85.6 | 73.5 | 84.0 |
| ConvNeXt (Liu et al., 2022) | 84.7 | 84.4 | 90.0 | 80.3 | 84.9 | 92.5 | 87.0 | 84.3 | 99.9 | 92.2 | 75.0 | 88.5 | 89.5 | 75.3 | 82.4 | 59.5 | 84.3 | 71.9 | 81.9 |
| Recce (Cao et al., 2022) | 68.9 | 54.1 | 51.0 | 55.4 | 57.4 | 83.2 | 71.2 | 70.2 | 94.2 | 67.9 | 74.4 | 76.9 | 59.6 | 55.2 | 57.4 | 62.1 | 66.5 | 64.3 | 64.0 |
| Sia (Sun et al., 2022) | 67.4 | 53.7 | 81.0 | 68.1 | 67.6 | 86.3 | 74.4 | 74.9 | 96.7 | 77.7 | 81.9 | 82.0 | 79.3 | 55.9 | 67.6 | 58.8 | 89.1 | 74.0 | 72.8 |
| IML-ViT (Ma et al., 2023) | 76.0 | 78.8 | 85.0 | 71.1 | 77.7 | 87.7 | 77.3 | 72.2 | 99.9 | 82.3 | 81.8 | 83.5 | 88.8 | 75.6 | 82.2 | 52.7 | 87.7 | 70.2 | 78.4 |
| Trufor (Guillaro et al., 2023) | 78.6 | 65.1 | 83.1 | 69.8 | 74.2 | 88.0 | 84.8 | 77.1 | 99.2 | 88.8 | 81.0 | 86.5 | 90.7 | 78.2 | 84.5 | 51.8 | 90.5 | 71.2 | 79.1 |
| UnivFD (Ojha et al., 2023) | 88.5 | 94.2 | 69.8 | 73.7 | 81.6 | 91.5 | 69.7 | 78.3 | 98.8 | 66.7 | 87.1 | 82.0 | 68.2 | 84.6 | 76.4 | 33.0 | 79.2 | 56.1 | 74.0 |
| FFDN (Chen et al., 2024) | 76.2 | 51.5 | 87.2 | 76.3 | 72.8 | 95.9 | 92.9 | 94.9 | 100. | 93.3 | 74.8 | 92.0 | 93.3 | 80.2 | 86.8 | 56.1 | 84.7 | 70.4 | 80.5 |
| Effort (Yan et al., 2024a) | 94.7 | 96.4 | 87.3 | 91.3 | **92.4** | 95.9 | 79.3 | 88.2 | 99.3 | 76.8 | 88.7 | 88.0 | 89.0 | 91.0 | **90.0** | 48.3 | 83.6 | 66.0 | 84.1 |
| Mesorch (Zhu et al., 2025a) | 83.0 | 72.7 | 87.1 | 78.8 | 80.4 | 88.2 | 80.1 | 78.3 | 99.6 | 87.5 | 82.5 | 86.0 | 87.6 | 76.0 | 81.8 | 58.9 | 88.5 | 73.7 | 80.5 |
| CO-SPY (Cheng et al., 2025) | 80.9 | 62.3 | 85.1 | 71.4 | 74.9 | 92.4 | 88.8 | 85.5 | 98.9 | 94.1 | 70.2 | 88.3 | 84.4 | 68.0 | 76.2 | 52.2 | 85.7 | 69.0 | 77.1 |
| CLIP-FFT | 81.8 | 66.9 | 85.7 | 74.9 | 77.3 | 95.4 | 90.4 | 88.8 | 100. | 93.1 | 76.8 | 90.8 | 87.8 | 70.8 | 79.3 | 56.9 | 89.2 | 73.1 | 80.1 |
| **SICA (Ours)** | 95.9 | 87.1 | 91.5 | 92.5 | 91.8 | 99.2 | 98.6 | 95.5 | 100. | 97.1 | 77.3 | **94.6** | 90.7 | 86.4 | 88.6 | 62.9 | 87.6 | **75.3** | 87.5 |

*Table 7.* **AP results on *OpenMMSec*.** The overall average is the macro-average of the domain averages. The best and second-best results are highlighted in **bold** and underlined, respectively.

| Method | Deepfake | | | | | AIGC | | | | | | | IMDL | | | Doc | | | Avg |
|---|---|---|---|---|---|---|---|---|---|---|---|---|---|---|---|---|---|---|---|
| | EFS | FE | FR | FS | Avg | GAN | Lat-Diff | Pix-Diff | AR | Comm | Other | Avg | Gen | Non-Gen | Avg | AIGC | Non-AIGC | Avg | |
| Resnet (He et al., 2016) | 72.4 | 41.5 | 67.7 | 56.9 | 59.6 | 65.6 | 52.5 | 15.1 | 84.1 | 25.8 | 29.6 | 45.5 | 83.6 | 41.8 | 62.7 | 18.3 | 97.3 | 57.8 | 56.4 |
| EfficientNet (Tan & Le, 2019) | 46.0 | 21.2 | 53.6 | 52.1 | 43.2 | 24.2 | 43.3 | 10.4 | 8.6 | 17.5 | 17.2 | 20.2 | 39.1 | 22.9 | 31.0 | 15.8 | 92.8 | 54.3 | 37.2 |
| CapsuleNet (Nguyen et al., 2019) | 61.1 | 41.0 | 68.2 | 52.6 | 55.7 | 55.8 | 48.8 | 12.2 | 28.9 | 19.6 | 31.7 | 32.8 | 81.0 | 39.0 | 60.0 | 14.2 | 96.6 | 55.4 | 51.0 |
| SegFormer (Xie et al., 2021) | 87.0 | 60.4 | 88.7 | 76.5 | 78.2 | 75.1 | 83.2 | 40.3 | 95.4 | 76.8 | 33.4 | 67.4 | 87.4 | 52.9 | 70.1 | 22.4 | 97.6 | 60.0 | 68.9 |
| Swin (Liu et al., 2021b) | 89.7 | 63.8 | 86.0 | 70.7 | 77.5 | 80.9 | 87.4 | 46.1 | 97.0 | 79.1 | 33.4 | 70.6 | 90.1 | 56.9 | 73.5 | 23.3 | 97.1 | 60.2 | 70.5 |
| ConvNeXt (Liu et al., 2022) | 83.2 | 66.9 | 89.5 | 74.6 | 78.5 | 75.8 | 82.0 | 33.1 | 96.9 | 79.4 | 29.7 | 66.2 | 87.7 | 52.4 | 70.0 | 23.1 | 96.8 | 60.0 | 68.7 |
| Recce (Cao et al., 2022) | 63.6 | 37.2 | 50.5 | 50.1 | 50.3 | 63.5 | 60.1 | 17.8 | 69.6 | 31.9 | 35.1 | 46.3 | 47.4 | 27.1 | 37.3 | 22.6 | 93.3 | 57.9 | 48.0 |
| Sia (Sun et al., 2022) | 59.7 | 32.5 | 77.3 | 58.7 | 57.0 | 63.2 | 60.6 | 15.4 | 74.1 | 39.6 | 36.6 | 49.1 | 72.0 | 26.9 | 49.5 | 21.8 | 97.9 | 59.8 | 53.9 |
| IML-ViT (Ma et al., 2023) | 68.3 | 57.4 | 78.4 | 59.7 | 65.9 | 70.4 | 63.9 | 16.9 | 98.0 | 50.1 | 37.5 | 56.1 | 84.9 | 47.9 | 66.4 | 17.8 | 97.6 | 57.7 | 61.5 |
| Trufor (Guillaro et al., 2023) | 72.9 | 36.3 | 78.3 | 58.8 | 61.6 | 66.8 | 77.2 | 25.9 | 70.7 | 69.4 | 32.8 | 57.1 | 86.7 | 47.9 | 67.3 | 17.2 | 98.0 | 57.6 | 60.9 |
| UnivFD (Ojha et al., 2023) | 87.7 | 88.2 | 68.9 | 72.1 | 79.2 | 75.3 | 60.3 | 25.9 | 83.8 | 24.3 | 58.7 | 54.7 | 54.8 | 68.5 | 61.6 | 12.9 | 94.5 | 53.7 | 62.3 |
| FFDN (Chen et al., 2024) | 74.0 | 43.9 | 85.7 | 69.2 | 68.2 | 86.8 | 94.1 | 69.8 | 99.9 | 91.4 | 42.4 | 80.7 | 91.2 | 57.1 | 74.2 | 19.8 | 97.0 | 58.4 | 70.4 |
| Effort (Yan et al., 2024a) | 94.1 | 93.3 | 87.9 | 91.6 | **91.7** | 87.1 | 67.3 | 48.4 | 91.0 | 30.5 | 48.0 | 62.0 | 85.8 | 78.7 | **82.2** | 16.8 | 96.2 | 56.5 | 73.1 |
| Mesorch (Zhu et al., 2025a) | 76.1 | 53.2 | 86.0 | 69.3 | 70.9 | 68.5 | 72.3 | 21.1 | 92.8 | 66.4 | 42.2 | 61.1 | 84.1 | 47.8 | 65.9 | 21.6 | 97.8 | 59.7 | 64.4 |
| CO-SPY (Cheng et al., 2025) | 79.4 | 45.8 | 86.1 | 65.3 | 69.2 | 78.5 | 85.9 | 47.1 | 92.4 | 81.9 | 25.6 | 68.6 | 81.7 | 40.6 | 61.1 | 19.2 | 97.2 | 58.2 | 64.3 |
| CLIP-FFT | 77.9 | 49.8 | 82.1 | 65.8 | 68.9 | 84.6 | 86.6 | 46.2 | 99.9 | 81.0 | 30.8 | 71.5 | 86.6 | 46.7 | 66.6 | 20.7 | 97.9 | 59.3 | 66.6 |
| **SICA (Ours)** | 95.5 | 82.1 | 92.4 | 92.2 | 90.5 | 97.0 | 97.9 | 77.3 | 99.7 | 91.2 | 34.9 | **83.0** | 89.4 | 70.8 | 80.1 | 27.1 | 97.5 | **62.3** | 79.0 |

methods. The dataset is designed to advance the evaluation of detection models' generalization under data distributions that more closely resemble real-world deployment scenarios.

**Celeb-DF-v2** (Li et al., 2020c) is designed to provide a higher-quality and more challenging deepfake data distribution. According to the official description, it contains 590 real videos and 5,639 corresponding DeepFake videos, with forgery quality approaching that of real-world online content, making it suitable for evaluating performance degradation and generalization of detectors under high-quality forgeries.

**DF40** (Yan et al., 2024b) covers 40 different deepfake techniques and provides multiple evaluation protocols along with large-scale comparative experiments to systematically analyze how dataset design and evaluation protocols affect generalization conclusions.

### D.3.2. AIGC

**DiffusionForensics** (Wang et al., 2023) is a benchmark for detecting diffusion-generated images introduced alongside the DIRE method, designed to evaluate the separability between real images and images generated by diffusion models. The paper presents it as a comprehensive benchmark for diffusion-based generation, aiming to assess the robustness and generalization of detectors to artifacts produced by different diffusion models.

**GenImage** (Zhu et al., 2023) is a million-scale AIGC image detection benchmark, described in the paper as containing over one million pairs of AI-generated and real images, covering diverse categories and synthesized by multiple advanced generators, including diffusion models and GANs. The authors propose evaluation settings such as cross-generator generalization and robustness to degradations to better reflect detection under unknown generators and image degradations in real-world scenarios.

*Table 8.* **F1 results on *OpenMMSec*.** The overall average is the macro-average of the domain averages. The best and second-best results are highlighted in **bold** and underlined, respectively.

| Method | Deepfake | | | | | AIGC | | | | | | | IMDL | | | Doc | | | Avg |
|---|---|---|---|---|---|---|---|---|---|---|---|---|---|---|---|---|---|---|---|
| | EFS | FE | FR | FS | Avg | GAN | Lat-Diff | Pix-Diff | AR | Comm | Other | Avg | Gen | Non-Gen | Avg | AIGC | Non-AIGC | Avg | |
| Resnet (He et al., 2016) | 66.5 | 33.2 | 68.4 | 59.1 | 56.8 | 47.4 | 48.2 | 24.5 | 20.1 | 31.5 | 37.5 | 34.9 | 75.0 | 46.6 | 60.8 | 20.9 | 88.5 | 54.7 | 51.8 |
| EfficientNet (Tan & Le, 2019) | 56.4 | 31.6 | 63.4 | 60.8 | 53.1 | 35.7 | 47.3 | 17.7 | 12.8 | 27.7 | 29.2 | 28.4 | 44.8 | 31.5 | 38.2 | 24.9 | 85.7 | 55.3 | 43.7 |
| CapsuleNet (Nguyen et al., 2019) | 57.5 | 26.5 | 66.1 | 54.5 | 51.1 | 46.7 | 39.5 | 19.7 | 19.8 | 22.5 | 38.1 | 31.0 | 69.9 | 36.6 | 53.2 | 12.6 | 90.7 | 51.7 | 46.8 |
| SegFormer (Xie et al., 2021) | 79.4 | 52.9 | 81.9 | 68.4 | 70.7 | 61.5 | 72.0 | 42.1 | 35.2 | 65.2 | 31.7 | 51.3 | 75.0 | 46.3 | 60.7 | 26.8 | 87.3 | 57.1 | 59.9 |
| Swin (Liu et al., 2021b) | 80.1 | 58.1 | 78.9 | 63.9 | 70.2 | 63.6 | 78.1 | 43.5 | 31.1 | 65.1 | 34.6 | 52.7 | 78.8 | 51.0 | 64.9 | 27.9 | 85.2 | 56.5 | 61.1 |
| ConvNeXt (Liu et al., 2022) | 72.8 | 59.9 | 81.9 | 62.1 | 69.1 | 64.0 | 68.7 | 36.5 | 41.1 | 68.5 | 27.7 | 51.1 | 75.8 | 46.0 | 60.9 | 27.7 | 83.9 | 55.8 | 59.2 |
| Recce (Cao et al., 2022) | 57.9 | 29.0 | 49.1 | 52.5 | 47.1 | 55.8 | 21.4 | 14.9 | 59.5 | 20.9 | 27.2 | 33.3 | 6.4 | 3.5 | 4.9 | 7.3 | 61.8 | 34.5 | 30.0 |
| Sia (Sun et al., 2022) | 55.4 | 31.4 | 73.1 | 58.6 | 54.6 | 49.9 | 50.7 | 23.5 | 24.8 | 42.5 | 42.0 | 38.9 | 66.0 | 31.7 | 48.8 | 25.7 | 89.8 | 57.7 | 50.0 |
| IML-ViT (Ma et al., 2023) | 63.3 | 45.7 | 76.6 | 64.8 | 62.6 | 52.6 | 53.3 | 24.0 | 28.6 | 47.8 | 38.6 | 40.8 | 72.1 | 41.9 | 57.0 | 11.7 | 86.0 | 48.9 | 52.3 |
| Trufor (Guillaro et al., 2023) | 66.0 | 39.3 | 75.2 | 55.1 | 58.9 | 54.5 | 68.6 | 33.0 | 28.5 | 58.1 | 35.6 | 46.4 | 74.7 | 40.8 | 57.7 | 18.7 | 91.9 | 55.3 | 54.6 |
| UnivFD (Ojha et al., 2023) | 63.5 | 40.7 | 65.6 | 64.9 | 58.7 | 58.2 | 48.8 | 29.8 | 26.8 | 29.6 | 51.1 | 40.7 | 51.5 | 59.5 | 55.5 | 18.9 | 92.2 | 55.5 | 52.6 |
| FFDN (Chen et al., 2024) | 62.5 | 35.5 | 79.2 | 59.0 | 59.1 | 77.2 | 84.3 | 64.4 | 58.7 | 82.3 | 30.9 | 66.3 | 75.4 | 32.7 | 54.0 | 24.0 | 83.8 | 53.9 | 58.3 |
| Effort (Yan et al., 2024a) | 84.4 | 73.6 | 78.3 | 83.2 | 79.9 | 65.0 | 54.7 | 41.7 | 28.8 | 33.9 | 52.0 | 46.0 | 74.5 | 71.0 | 72.8 | 18.8 | 86.1 | 52.4 | 62.8 |
| Mesorch (Zhu et al., 2025a) | 72.4 | 45.5 | 79.7 | 70.0 | 66.9 | 53.1 | 60.9 | 29.5 | 27.7 | 56.2 | 41.4 | 44.8 | 72.4 | 44.7 | 58.6 | 26.6 | 88.6 | 57.6 | 57.0 |
| CO-SPY (Cheng et al., 2025) | 68.6 | 34.8 | 75.4 | 55.8 | 58.6 | 58.6 | 73.2 | 38.1 | 28.2 | 62.9 | 23.3 | 47.4 | 67.3 | 32.5 | 49.9 | 25.7 | 86.0 | 55.9 | 53.0 |
| CLIP-FFT | 67.4 | 37.9 | 77.1 | 53.8 | 59.1 | 72.5 | 71.3 | 44.4 | 53.2 | 71.9 | 24.5 | 56.3 | 74.3 | 34.1 | 54.2 | 25.2 | 88.1 | 56.7 | 56.6 |
| **SICA (Ours)** | 88.9 | 73.5 | 84.6 | 83.9 | **82.7** | 87.6 | 92.4 | 68.7 | 62.7 | 82.8 | 27.6 | **70.3** | 77.7 | 59.3 | 68.5 | 28.8 | 86.0 | 57.4 | **69.7** |

**CommunityForensics** (Park & Owens, 2025) is centered on large-scale generator diversity: the authors systematically download and sample thousands of text-to-image diffusion models, supplemented with images from various open-source and commercial generators, resulting in a dataset of 2.7M images from 4,803 models. The paper empirically demonstrates that model diversity and distribution coverage in training data are critical for generalizing detection to unseen generators.

### D.3.3. IMDL

**CASIAv2** (Dong et al., 2013) is one of the most widely used traditional image manipulation detection datasets, commonly employed for studying the detection and localization of splicing and copy-move forgeries.

**IMD2020** (Novozamsky et al., 2020) is designed for manipulation detection under real internet-sourced conditions, containing 2,010 genuinely manipulated images along with their corresponding authentic versions for comparison. It is commonly used to evaluate a method's adaptability to real-world manipulation types and acquisition noise.

**tamperCOCO** (Kwon et al., 2022) is an image manipulation detection and localization dataset constructed in the CAT-Net work based on COCO (**?**). Its core idea is to synthesize manipulated images from real COCO images using an automated manipulation pipeline, while retaining pixel-level annotations of the manipulated regions for training and evaluating image manipulation detection and localization models.

**MIML** (Qu et al., 2024) is a large-scale modern image manipulation localization dataset, containing 123,150 manually forged images with pixel-level annotations. It is designed to improve the generalization of localization models under modern editing styles and pipelines, and introduces quality assessment mechanisms to ensure annotation reliability.

**Autosplice** (Jia et al., 2023) targets the risks of text-prompt–driven generative editing by using DALL·E 2 to generate local image regions conditioned on text prompts and automatically splice them into images, constructing a dataset of 5,894 real/manipulated image pairs. The paper defines both detection and localization tasks and reports that many existing forensic models suffer significant performance degradation on previously unseen prompt-driven edits.

**GRE** (Sun et al., 2024) is a large-scale dataset for generative region-editing detection. Both the paper and the official repository position it as addressing the forensic gap in the era of generative editing, reporting a scale of over 228K edited images and covering diverse editing methods with different characteristics, designed to systematically evaluate and advance detection approaches for this setting.

**OpenSDI** (Wang et al., 2025) is an open-world diffusion-generated image identification dataset designed to evaluate methods under open-world and unknown-distribution diffusion generation scenarios.

### D.3.4. Doc

**Doctamper** (Qu et al., 2023) is a captured document image text manipulation dataset covering documents such as contracts, receipts, invoices, and books. The tampering types include copy-move, splicing, and print-based edits.

**OSTF** (Qu et al., 2025) is a dataset for open-set text manipulation detection, containing text manipulations generated by

*Table 9.* **AUC results on *IFF-Protocol* proposed in ForensicHub (Du et al., 2025)**. We use the values reported in ForensicHub. The best and second-best results are highlighted in **bold** and underlined, respectively.

| Method | Deepfake | | | IMDL | | | AIGC | | Doc | | | Avg |
|---|---|---|---|---|---|---|---|---|---|---|---|---|
| | FF-c40 | CDFv2 | DFD | Columbia | IMD2020 | Autosplice | DF | GenImage | T-SROIE | OSTF | RTM | |
| Resnet | 0.681 | 0.730 | 0.793 | 0.482 | 0.533 | 0.738 | 0.619 | 0.797 | 0.951 | 0.681 | 0.662 | 0.697 |
| Xception | 0.728 | 0.719 | 0.870 | 0.465 | 0.537 | 0.756 | 0.757 | 0.980 | 0.966 | 0.762 | 0.734 | 0.752 |
| EfficientNet | 0.504 | 0.535 | 0.517 | 0.623 | 0.506 | 0.483 | 0.544 | 0.597 | 0.884 | 0.581 | 0.512 | 0.571 |
| Segformer | 0.691 | 0.748 | 0.862 | 0.409 | 0.562 | 0.824 | 0.805 | 0.998 | 0.980 | 0.866 | 0.736 | 0.771 |
| Swin | 0.771 | 0.746 | 0.901 | 0.636 | 0.631 | 0.864 | 0.915 | 0.999 | 0.990 | 0.856 | 0.758 | 0.824 |
| ConvNext | 0.794 | 0.784 | 0.911 | 0.625 | 0.598 | 0.825 | 0.895 | 1.000 | 0.994 | 0.849 | 0.762 | 0.822 |
| Capsule-Net | 0.613 | 0.660 | 0.699 | 0.330 | 0.527 | 0.745 | 0.546 | 0.971 | 0.946 | 0.704 | 0.670 | 0.674 |
| RECCE | 0.634 | 0.602 | 0.727 | 0.506 | 0.492 | 0.642 | 0.684 | 0.906 | 0.542 | 0.688 | 0.555 | 0.634 |
| SPSL | 0.730 | 0.726 | 0.876 | 0.419 | 0.545 | 0.759 | 0.770 | 0.987 | 0.972 | 0.769 | 0.738 | 0.754 |
| Sia | 0.629 | 0.584 | 0.671 | 0.653 | 0.483 | 0.626 | 0.593 | 0.748 | 0.610 | 0.677 | 0.574 | 0.622 |
| Effort | 0.805 | 0.846 | 0.930 | 0.979 | 0.861 | 0.943 | 0.930 | 0.992 | 0.960 | 0.834 | 0.732 | 0.892 |
| MVSS-Net | 0.713 | 0.700 | 0.857 | 0.298 | 0.539 | 0.795 | 0.671 | 0.994 | 0.978 | 0.806 | 0.741 | 0.736 |
| Trufor | 0.642 | 0.698 | 0.832 | 0.306 | 0.564 | 0.808 | 0.726 | 0.996 | 0.979 | 0.805 | 0.732 | 0.735 |
| IML-ViT | 0.750 | 0.726 | 0.851 | 0.483 | 0.556 | 0.819 | 0.627 | 0.991 | 0.972 | 0.800 | 0.703 | 0.753 |
| Mesorch | 0.767 | 0.814 | 0.867 | 0.285 | 0.570 | 0.773 | 0.629 | 0.996 | 0.982 | 0.819 | 0.739 | 0.749 |
| DualNet | 0.637 | 0.552 | 0.540 | 0.268 | 0.517 | 0.748 | 0.899 | 0.988 | 0.935 | 0.658 | 0.657 | 0.673 |
| HiFiNet | 0.587 | 0.611 | 0.648 | 0.745 | 0.534 | 0.677 | 0.575 | 0.756 | 0.937 | 0.663 | 0.615 | 0.668 |
| UnivFD | 0.690 | 0.671 | 0.798 | 0.886 | 0.786 | 0.785 | 0.742 | 0.813 | 0.938 | 0.684 | 0.569 | 0.760 |
| FatFormer | 0.842 | 0.770 | 0.866 | 0.199 | 0.585 | 0.784 | 0.941 | 0.999 | 0.983 | 0.806 | 0.751 | 0.758 |
| CO-SPY | 0.819 | 0.780 | 0.875 | 0.460 | 0.716 | 0.779 | 0.940 | 0.989 | 0.969 | 0.836 | 0.748 | 0.829 |
| DTD | 0.498 | 0.520 | 0.490 | 0.679 | 0.498 | 0.506 | 0.457 | 0.499 | 0.748 | 0.595 | 0.496 | 0.544 |
| FFDN | 0.714 | 0.699 | 0.871 | 0.553 | 0.624 | 0.927 | 0.999 | 1.000 | 0.997 | 0.893 | 0.782 | 0.824 |
| **SICA (Ours)** | 0.825 | 0.862 | 0.939 | 0.965 | 0.883 | 0.950 | 0.929 | 0.999 | 0.985 | 0.897 | 0.761 | **0.910** |

eight different AIGC text editing models.

**RTM** (Luo et al., 2025) is a dataset proposed to address the scarcity of real-world text manipulation data. It covers a wide range of manipulation types, including copy-move, splicing, print, and erasure, across diverse document types such as scanned forms.

## D.4. Domain SoTAs Involved in Our Work

### D.4.1. DEEPFAKE

**CapsuleNet** (Nguyen et al., 2019) applies capsule networks to digital forensics, aiming to detect forged images and videos across diverse scenarios (e.g., replay attacks and CNN-generated fakes), leveraging capsule routing agreement to improve detection on challenging forgeries.

**Recce** (Cao et al., 2022) is an end-to-end reconstruction–classification learning framework for face forgery detection, where reconstruction learning over genuine faces helps mine common real-face features, and reconstruction discrepancies serve as cues that are jointly optimized with classification to better separate real and fake faces.

**Sia** (Sun et al., 2022) is a forgery detection approach that introduces a self-information metric into attention, proposing a plug-and-play Self-Information Attention module that emphasizes informative regions and recalibrates feature responses to improve detection performance and generalization.

### D.4.2. AIGC

**UnivFD** (Ojha et al., 2023) shows that standard trained real-vs-fake classifiers generalize poorly to new generative models and proposes learning-free fake detection by operating in a pretrained vision–language feature space, using simple nearest-neighbor or linear probing to achieve strong generalization to unseen diffusion and autoregressive models.

**Effort** (Yan et al., 2024a) identifies an "asymmetry phenomenon" in AI-generated image detection where models overfit limited fake patterns and become low-rank, and proposes SVD-based orthogonal subspace decomposition: freezing principal components to preserve pretrained knowledge while adapting remaining components to learn forgery patterns for better

*Table 10.* **AUC results on Deepfake benchmark.** We train the model on FaceForensics++ (c23) (Rossler et al., 2019) and evaluate it on other test sets. Method results marked with * are our re-implementations, while the remaining values are taken from DeepfakeBench (Yan et al., 2023). The best and second-best results are highlighted in **bold** and underlined, respectively.

| Method | Within Domain | | | | | | Cross Domain | | | | | | | |
|---|---|---|---|---|---|---|---|---|---|---|---|---|---|---|
| | FF-c40 | FF-DF | FF-F2F | FF-FS | FF-NT | Avg | CDFv1 | CDFv2 | DFD | DFDC | DFDCP | FaceShifter | UADFV | Avg |
| Meso4 | 0.5920 | 0.6771 | 0.6170 | 0.5946 | 0.5701 | 0.6102 | 0.7358 | 0.6091 | 0.5481 | 0.5560 | 0.5994 | 0.5660 | 0.7150 | 0.6185 |
| MesoIncep | 0.7278 | 0.8542 | 0.8087 | 0.7421 | 0.6517 | 0.7569 | 0.7366 | 0.6966 | 0.6069 | 0.6226 | 0.7561 | 0.6438 | 0.9049 | 0.7096 |
| CNN-Aug | 0.7846 | 0.9048 | 0.8788 | 0.9026 | 0.7313 | 0.8404 | 0.7420 | 0.7027 | 0.6464 | 0.6361 | 0.6170 | 0.5985 | 0.8739 | 0.6881 |
| Xception | 0.8261 | 0.9799 | 0.9785 | 0.9833 | 0.9385 | 0.9413 | 0.7794 | 0.7365 | 0.8163 | 0.7077 | 0.7374 | 0.6249 | 0.9379 | 0.7629 |
| EfficientB4 | 0.8150 | 0.9757 | 0.9758 | 0.9797 | 0.9308 | 0.9354 | 0.7909 | 0.7487 | 0.8148 | 0.6955 | 0.7283 | 0.6162 | 0.9472 | 0.7631 |
| Capsule | 0.7040 | 0.8669 | 0.8634 | 0.8734 | 0.7804 | 0.8176 | 0.7909 | 0.7472 | 0.6841 | 0.6465 | 0.6568 | 0.6465 | 0.9078 | 0.7257 |
| FWA | 0.7357 | 0.9210 | 0.9000 | 0.8843 | 0.8120 | 0.8506 | 0.7897 | 0.6680 | 0.7403 | 0.6132 | 0.6375 | 0.5551 | 0.8539 | 0.6940 |
| X-ray | 0.7925 | 0.9794 | 0.9872 | 0.9871 | 0.9290 | 0.9350 | 0.7093 | 0.6786 | 0.7655 | 0.6326 | 0.6942 | 0.6553 | 0.8989 | 0.7192 |
| FFD | 0.8237 | 0.9803 | 0.9784 | 0.9853 | 0.9306 | 0.9397 | 0.7840 | 0.7435 | 0.8024 | 0.7029 | 0.7426 | 0.6056 | 0.9450 | 0.7609 |
| CORE | 0.8194 | 0.9787 | 0.9803 | 0.9823 | 0.9339 | 0.9390 | 0.7798 | 0.7428 | 0.8018 | 0.7049 | 0.7341 | 0.6032 | 0.9412 | 0.7583 |
| Recce | 0.8190 | 0.9797 | 0.9779 | 0.9785 | 0.9357 | 0.9382 | 0.7677 | 0.7319 | 0.8119 | 0.7133 | 0.7419 | 0.6095 | 0.9446 | 0.7601 |
| UCF | 0.8399 | 0.9883 | 0.9840 | 0.9896 | 0.9441 | 0.9492 | 0.7793 | 0.7527 | 0.8074 | 0.7191 | 0.7594 | 0.6462 | 0.9528 | 0.7738 |
| F3Net | 0.8271 | 0.9793 | 0.9796 | 0.9844 | 0.9354 | 0.9412 | 0.7769 | 0.7352 | 0.7975 | 0.7021 | 0.7354 | 0.5914 | 0.9347 | 0.7533 |
| SPSL | 0.8174 | 0.9781 | 0.9754 | 0.9829 | 0.9299 | 0.9367 | 0.8150 | 0.7650 | 0.8122 | 0.7040 | 0.7408 | 0.6437 | 0.9424 | 0.7747 |
| SRM | 0.8114 | 0.9733 | 0.9696 | 0.9740 | 0.9295 | 0.9316 | 0.7926 | 0.7552 | 0.8120 | 0.6995 | 0.7408 | 0.6014 | 0.9427 | 0.7635 |
| Sia* | 0.7367 | 0.9621 | 0.9492 | 0.9537 | 0.9143 | 0.9032 | 0.7298 | 0.7424 | 0.8102 | 0.7236 | 0.7834 | 0.6608 | 0.7270 | 0.7396 |
| Effort* | 0.8062 | 0.9928 | 0.9839 | 0.9806 | 0.9625 | 0.9452 | 0.8822 | 0.8772 | 0.9438 | 0.8194 | 0.8711 | 0.7981 | 0.9702 | 0.8803 |
| **SICA (Ours)** | 0.8344 | 0.9946 | 0.9866 | 0.9855 | 0.9739 | **0.9550** | 0.9067 | 0.9068 | 0.9436 | 0.8241 | 0.8763 | 0.8302 | 0.9740 | **0.8945** |

generalization.

*Table 11.* **Accuracy results on AIGC benchmark.** We adopt GenImage (Zhu et al., 2023) as the benchmark, training only on data generated by Stable Diffusion v1.4 (Rombach et al., 2022) and testing on other generators. We use the value reported in GenImage. The best and second-best results are highlighted in **bold** and underlined, respectively.

| Method | Midjourney | SD V1.4 | SD V1.5 | ADM | GLIDE | Wukong | VQDM | BigGAN | Avg |
|---|---|---|---|---|---|---|---|---|---|
| ResNet-50 | 54.9 | 99.9 | 99.7 | 53.5 | 61.9 | 98.2 | 56.6 | 52.0 | 72.1 |
| DeiT-S | 55.6 | 99.9 | 99.8 | 49.8 | 58.1 | 98.9 | 56.9 | 53.5 | 71.6 |
| Swin-T | 62.1 | 99.9 | 99.8 | 49.8 | 67.6 | 99.1 | 62.3 | 57.6 | 74.8 |
| CNNSpot | 52.8 | 96.3 | 95.9 | 50.1 | 39.8 | 78.6 | 53.4 | 46.8 | 64.2 |
| Spec | 52.0 | 99.4 | 99.2 | 49.7 | 49.8 | 94.8 | 55.6 | 49.8 | 68.8 |
| F3Net | 50.1 | 99.9 | 99.9 | 49.9 | 50.0 | 99.9 | 49.9 | 49.9 | 68.7 |
| GramNet | 54.2 | 99.2 | 99.1 | 50.3 | 54.6 | 98.9 | 50.8 | 51.7 | 69.9 |
| **SICA (Ours)** | 65.5 | 100.0 | 99.9 | 51.1 | 66.1 | 99.7 | 64.4 | 55.6 | **75.3** |

**CO-SPY** (Cheng et al., 2025) addresses generalization and post-processing robustness (e.g., JPEG) in synthetic image detection by enhancing semantic features (e.g., semantic anomalies) and pixel/artifact features (e.g., low-level differences) separately and then adaptively integrating them.

D.4.3. IMDL

**IML-ViT** (Ma et al., 2023) is a ViT-based benchmark paradigm for image manipulation localization, motivated by the need to capture non-semantic discrepancies via explicit comparisons between manipulated and authentic regions, and designed with high-resolution capacity, multi-scale features, and edge supervision to work with limited data.

**Trufor** (Guillaro et al., 2023) is a forensic framework for trustworthy image forgery detection and localization that fuses high-level and low-level traces through a transformer-based architecture combining RGB content with a learned noise-sensitive fingerprint.

**Mesorch** (Zhu et al., 2025a) introduces a mesoscopic perspective for manipulation localization by integrating microscopic traces and macroscopic semantic changes, using parallel CNNs (micro details) and Transformers (macro information) with multi-scale orchestration to improve performance, efficiency, and robustness.

### D.4.4. Doc

**FFDN** (Chen et al., 2024) targets document image tampering detection by jointly modeling spatial and frequency cues: a Visual Enhancement Module makes subtle traces more visible, while a Wavelet-like Frequency Enhancement explicitly decomposes and preserves high-frequency details crucial for detecting weak tampering artifacts.

### D.5. Discussion on the Real-Fake Imbalance in Documents

The discrepancy of the Doc domain in our dataset inherently stems from the current landscape of public Doc tampering datasets (Qu et al., 2023; Wang et al., 2022c). Specifically, due to strict privacy constraints and copyright issues associated with authentic documents (e.g., personal IDs, financial invoices, and contracts), prior works predominantly provide tampered images with very few corresponding authentic counterparts. While we carefully sourced diverse authentic documents to maximize coverage, achieving a strict 1:1 real-to-fake ratio was constrained by source availability. Importantly, similar real-to-fake imbalances are prevalent in established forensic benchmarks (e.g., standard Deepfake datasets (Yan et al., 2024b; Rossler et al., 2019)) and do not compromise the fairness or validity of the evaluation.

## E. Additional Experiments

### E.1. Implementation Details

We insert LoRA into all linear layers within the attention modules of the ViT. The model is trained for 10 epochs with a batch size of 96 on four NVIDIA RTX Pro 6000 GPUs, using the AdamW optimizer (Loshchilov & Hutter, 2019). We use PyTorch (Paszke et al., 2019) for implementation. The learning rate follows a cosine decay (Loshchilov & Hutter, 2017) schedule from $1 \times 10^{-4}$ to $1 \times 10^{-5}$. Input images are resized to $224 \times 224$ to match the ViT input resolution. We use only the binary cross-entropy (BCE) loss. For the compared detectors, we use the default configurations provided in ForensicHub (Du et al., 2025) and train them on *OpenMMSec* using the same protocol. All models are trained using the same data augmentation.

### E.2. AUC, AP and F1 Results of Benchmark

We also report AUC, AP, and F1 in Tables 6, 7, and 8, respectively.

### E.3. Experiments on ForensicHub Benchmark

Although the FID benchmark ForensicHub (Du et al., 2025) has limitations in diverse faking types, balanced data volume, and rich image sources, we still adopt its proposed *IFF-Protocol* to evaluate the performance of SICA. As shown in Tab. 9, SICA still demonstrates superior generalization. However, compared to our *OpenMMSec* benchmark, the *IFF-Protocol* exhibits evaluation bottlenecks, further highlighting the advantages of *OpenMMSec* in terms of diverse faking types, balanced data volume, and rich image sources.

### E.4. Experiments on Subdomain Benchmarks

We additionally evaluate SICA on subdomain benchmarks, using DeepfakeBench (Yan et al., 2023) for Deepfake detection and GenImage (Zhu et al., 2023) for AIGC detection. The results, reported in Tab. 10 and Tab. 11, show that SICA also achieves highly comparable performance within individual subdomains.

