# OpenReview forum: "Can We Build a Monolithic Model for Fake Image Detection? SICA: Semantic-Induced Constrained Adaptation for Unified-Yet-Discriminative Artifact Feature Space Reconstruction"
_ICML.cc/2026/Conference — ICML 2026 regular_

### Official Review · Reviewer_TRbs · 2026-03-04

**Soundness:** 3
**Presentation:** 3
**Significance:** 4
**Originality:** 4
**Overall Recommendation:** 5
**Confidence:** 4

**Summary:**

This paper proposes a monolithic paradigm for Fake Image Detection (FID), introducing Semantic-Induced Constrained Adaptation (SICA) to prevent the collapse of the artifact feature space when unifying heterogeneous domains. Furthermore, the authors present a comprehensive benchmark dataset, OpenMMSec, and demonstrate the effectiveness of their approach through extensive experiments.

**Compliance With Llm Reviewing Policy:**

Affirmed.

**Final Justification:**

This is generally a solid work, and during rebuttal the authors have addressed all my concerns. After reviewing their full discussion with other reviewers, I remain supportive of this work and recommend acceptance.

**Key Questions For Authors:**

All my concerns are listed in weakness.

**Limitations:**

yes

**Strengths And Weaknesses:**

**Strength**
1. The insights of this paper are clear and easy to follow.
2. While subdomains of forgery detection exist, the unification into a single Fake Image Detection (FID) task is a significant and necessary endeavor for real-world forensics. The perspective on the role of semantics in image forensics is novel.
3. The proposed dataset is comprehensive and well-designed, and the experiments provide substantial support for the “artifact feature space collapse” hypothesis.

**Weakness**
1. The related work section could be further expanded, as the targeted FID task is composed of four distinct subdomain tasks.
2. The Document domain in the proposed dataset is notably smaller and exhibits a real-to-fake imbalance compared to others. The authors should briefly explain this discrepancy.
3. Although the spectral analysis provides theoretical backing, adding visual interpretability (e.g., Grad-CAM) would make the claims more intuitive. Visualizing whether the model's attention correctly focuses on manipulation artifacts across different subdomains would further strengthen the paper.

---

> ### Author Rebuttal · Authors · 2026-03-30
>
> We greatly appreciate your careful reading of our work and the thoughtful suggestions you have offered. Please find our responses to your concerns below.
>
> - "The related work section could be further expanded, as the targeted FID task is composed of four distinct subdomain tasks."
>
> Thank you for this constructive suggestion. We agree that a more granular discussion of the four subdomains will significantly enhance the completeness and contextual depth of our paper.
>
> In the revised final version manuscript, we will systematically expand the related work section to provide a comprehensive review of each individual subdomain. Specifically, we will detail their unique task characteristics and review the most representative state-of-the-art (SOTA) models within each area. Furthermore, we will summarize the detailed commonly exploited domain-specific artifacts (e.g., blending boundaries in face forgery vs. high-frequency noise in generative models) and their corresponding feature extraction strategies. By providing a detailed cross-domain comparison, we will explicitly highlight the intrinsic differences among these heterogeneous artifacts. This expanded discussion will better contextualize the fundamental motivation of our work: addressing the structural conflicts when unifying these diverse, domain-specific artifacts.
>
> - "The Document domain in the proposed dataset is notably smaller and exhibits a real-to-fake imbalance compared to others. The authors should briefly explain this discrepancy."
>
> We would like to clarify that this discrepancy inherently stems from the current landscape of public Doc tampering datasets [1,2]. Specifically, due to strict privacy constraints and copyright issues associated with authentic documents (e.g., personal IDs, financial invoices, and contracts), prior works predominantly provide tampered images with very few corresponding authentic counterparts. While we carefully sourced diverse authentic documents to maximize coverage, achieving a strict 1:1 real-to-fake ratio was constrained by source availability. Importantly, similar real-to-fake imbalances are prevalent in established forensic benchmarks (e.g., standard Deepfake datasets [3,4]) and do not compromise the fairness or validity of the evaluation. As suggested, we will explicitly include a brief discussion detailing these dataset characteristics and constraints in the revised manuscript.
>
> [1] Towards robust tampered text detection in document image: New dataset and new solution, CVPR23
>
> [2] Tampered text detection via RGB and frequency relationship modeling
>
> [3] Faceforensics++: Learning to detect manipulated facial images, ICCV19
>
> [4] DF40: Toward Next-Generation Deepfake Detection, NIPS24
>
> - "Although the spectral analysis provides theoretical backing, adding visual interpretability (e.g., Grad-CAM) would make the claims more intuitive. Visualizing whether the model's attention correctly focuses on manipulation artifacts across different subdomains would further strengthen the paper."
>
> Thank you for this constructive suggestion. To provide visual interpretability, we employed EigenCAM (which is well-suited for Vision Transformer architectures like our CLIP backbone) to extract and visualize SICA’s attention maps for fake images across all four distinct subdomains. The visualization results are provided in the anonymous link: https://anonymous.4open.science/r/some_pic-7037/attention_map.png.
>
> The visualizations confirm that SICA accurately localizes subdomain-specific artifacts in alignment with prior studies: For Deepfake, the attention is concentrated on facial region. For AIGC, SICA correctly captures the global anomalies and synthetic traces distributed across the entire image. For IMDL and Doc, SICA’s attention precisely localizes the tampered regions. This visual evidence directly corroborates our core claim: SICA successfully reconstructs the artifact feature space by accommodating domain-specific heterogeneous artifacts without cross-domain interference. We will prominently feature these qualitative results and the corresponding analysis in the revised manuscript to further strengthen the intuitiveness of our contributions.

---

> > ### Author Rebuttal · Reviewer_TRbs · 2026-04-02
> >
> > The authors have addressed all my concerns. This is generally a solid work. The findings of the heterogeneous phenomenon and feature space collapse are insightful and make sense. The SICA method also aligns with my observations. Besides, this paper is well-written, with clear motivations and illustrations. Therefore, I keep my final rating as 5 and suggest acceptance.

---

> > > ### Author Response · Authors · 2026-04-07
> > >
> > > We sincerely thank you for your consistent support and positive evaluation of our work. Your insightful comments have significantly helped us better highlight our core contributions, and we will ensure all your suggestions are reflected in the final version.

---

### Official Review · Reviewer_QUXG · 2026-03-13

**Soundness:** 2
**Presentation:** 3
**Significance:** 2
**Originality:** 3
**Overall Recommendation:** 3
**Confidence:** 5

**Summary:**

This paper aims to establish a unified fake image detection (FID) across four forensic subdomains. It identifies that the heterogeneous artifacts across subdomains result in feature collapse and degrade detection performance. The paper addresses this challenge by building a “unified-yet-discriminative” representation, which leverages semantic priors from a pretrained CLIP and learns forensic artifacts via LoRA. A new OpenMMSec dataset is constructed for benchmarking FID performance. The proposed approach outperforms 15 existing methods on OpenMMSec.

**Compliance With Llm Reviewing Policy:**

Affirmed.

**Final Justification:**

This paper has certain merits. It studies cross-domain interference in FID and proposes a simple approach based on fine-tuning CLIP with LoRA. The empirical observations and experiments provide some interesting insights.

However, a main concern remains that the central claim of  "feature collapse" is still supported primarily by indirect evidence rather than explicit representation-level quantitative analysis. Since this concept serves as a key motivation for the proposed method, more direct feature-space analysis would be important to substantiate the claim.

I acknowledge the authors' effort in clarifying several points during the rebuttal. I raise my score from 2 to 3.

**Key Questions For Authors:**

1. Please provide quantitative evidence that directly demonstrate feature collapse and show how SICA alleviates it.

2. Section 4.3 suggests that forensic artifacts may not be orthogonal to dominant semantic directions. Have you analyzed how the degree of entanglement varies across subdomains or data sources? Could FID benefit from explicitly modeling this entanglement?

3. How does SICA perform when images undergo common post-processing (e.g., compression, resizing, blur) or adversarial attacks intended to evade detection?

**Limitations:**

Partially. The paper presents several failure cases but simply suggests using stronger backbones to address the problem.
A discussion on how to improve the robustness on distorted images could be included.

**Strengths And Weaknesses:**

Strengths:
1. The paper identifies that building an effective monolithic FID model is hindered by feature space collapse caused by the heterogeneous nature of artifacts across subdomains.
2. Combining CLIP’s strong semantic prior with LoRA fine-tuning, the paper implements an effective monolithic FID model.
3. A large-scale FID dataset OpenMMSec is built, supplying valuable material for the FID community.

Weaknesses:
1. The proposed solution (CLIP + LoRA) is straightforward and offers limited novelty.
2. The attribution of poor monolithic performance to artifact feature collapse is only discussed qualitatively. Eq. (1) in Section 4.1 neither rigorously models the feature collapse (the union of features does not necessarily lead to collapse) nor directly supports the proposed method.
3. While the paper terms “reconstruction of the artifact feature space,” the proposed approach only adapts CLIP’s space to fake image detection via LoRA fine-tuning rather than actually reconstructing feature space.
4. Section 4.3 and the experimental results show that learning artifacts in the subspace orthogonal to dominant semantic directions (Effort) underperforms SICA. This implies that there are possible entanglements between semantics and forgery artifacts. However, this phenomenon is not analyzed in-depth. Would such entanglements depend on data sources?
5. In practice, images may be subjected to distortions or adversarial perturbations. The robustness of the proposed method under such conditions is not evaluated.
6. Although OpenMMSec covers 98 fine-grained faking types, the paper only mentions that 26/72 types are used for training/testing without detailed splits. This hinders a clear assessment of cross-type generalization.

---

> ### Author Rebuttal · Authors · 2026-03-30
>
> Thank you for taking the time to review our manuscript and for providing such insightful and constructive comments. We show our rebuttal in below.
>
> **First of all, with all of our respect, we can hardly agree that our solution is straightforward or offers limited novelty due to**:
>
> 1. As far as we know (also discussed in our paper from line 36-45), the FID still fully rely on ensemble methods and all previous efforts on seeking a monolithic FID solution result in failures. This proves that building a monolithic FID model is exceedingly difficult and thereby, SICA is not straightforward or intuitive at all to the researchers. **Otherwise, this prominent bottleneck would have been resolved long ago.**
> 2. Our primary novelty lies not in the architectural combination itself, **but in diagnosing the root cause of past failures, the artifact feature space collapse, and introducing a semantic-induced mechanism to actively reconstruct this space.** LoRA is used as a standard PEFT tool in subdomains[1,2], and these domain-specific models cannot be trivially transferred to the broader FID. Such direct applications inherently fail in unified FID without our structural insights to resolve the heterogeneous phenomenon, proving our underlying mechanism is obviously non-trivial.
>
> We may overlooked some literature that supports your "straightforward" assessment. **Would you mind pointing us to any prior work that either applies LoRA to unified cross-domain FID or formally diagnoses the heterogeneous artifact phenomenon**, so that we may fully address your concern.
>
> **Secondly, we would like to clarify your misplaced focus about "entanglements between semantics and forgery artifacts"**, which shifts the paper's focus away from our core objective. Our work focuses entirely on solving how artifacts should be represented in monolithic FID (i.e., preventing feature collapse), which is a fundamental prerequisite. Only after SICA works and we obtain a valid artifact feature space can the community even begin to meaningfully discuss whether these artifacts entangle with semantics. That is a highly complex, downstream challenge that warrants a dedicated, independent study. **Nevertheless, we are highly open to supplementing specific analyses dependencies if you have concrete experimental suggestions.**
>
> Please find our responses to the remaining concerns below.
>
> - Concerns regarding feature collapse and feature space reconstruction.
>
> The quantitative evidence of collapse lies in the performance gap between single-domain and unified multi-domain training (line 95 and Sec 5.3). As shown in Fig. 7 (left), baseline FFT suffers significant performance drops (large negative diagonals) under unified training, proving that forcing heterogeneous artifacts into a shared space causes severe interference and collapse.
>
> Conversely, SICA actively restructures artifact space. Sec 4.3 spectral analysis proves SICA explicitly learns artifact representations beyond CLIP's original semantic space. Furthermore, Sec 5.3 (Tab. 3, Fig. 7 right) demonstrates that SICA suffers almost zero degradation, proving the reconstruction.
>
> - "How does SICA perform when images undergo common post-processing or adversarial attacks"
>
> We randomly sampled 2,500 real and 2,500 fake images in OpenMMSec to evaluate SICA's robustness against two SOTAs across four post-processing methods: Gaussian Blur, Gaussian Noise, JPEG Compression, and Resize. The results, provided in the anonymous link (https://anonymous.4open.science/r/some_pic-7037/robust.png), demonstrate SICA's superior robustness under varying post-processing conditions.
>
> Furthermore, adversarial attacks are rarely considered a practical robustness metric in image forensics[3,4], as nearly all existing methods inherently fail against them. We evaluated SICA, Effort, and Trufor against Projected Gradient Descent (PGD) attacks, and the AUC for all three methods dropped to near zero. **Could you kindly direct us to any prior FID or subdomain work focusing on detection accuracy that evaluates adversarial robustness, so we may use it as a reference to further refine our experiments?**
>
> - "The paper only mentions that 26/72 types are used for training/testing without detailed splits."
>
> The 26 training types were carefully selected to ensure both domain balance and an optimal difficulty level for evaluating cross-type generalization, as too few or too many training types would lead to universally poor or saturated performance, obscuring meaningful model comparisons. We will explicitly detail the exact 26/72 fine-grained faking type splits in the final verison.
>
> [1] Adapt to Scarcity: Few-Shot Deepfake Detection via Low-Rank Adaptation
>
> [2] Deeclip: A robust and generalizable transformer-based framework for detecting ai-generated images
>
> [3] Deepfakebench: A comprehensive benchmark of deepfake detection, NIPS23
>
> [4] Imdl-benco: A comprehensive benchmark and codebase for image manipulation detection & localization, NIPS24

---

> > ### Author Rebuttal · Reviewer_QUXG · 2026-04-03
> >
> > Thank you for the rebuttal. Some of my concerns have been addressed, especially by clarifying the contribution beyond the CLIP+LoRA combination and adding post-processing robustness experiments.
> >
> > However, some key concerns are only partially addressed. If the authors could provide further clarification and analysis, the paper would become more convincing. Particularly:
> >
> > 1. The rebuttal mainly uses the performance gap between single-domain and unified multi-domain training as evidence of feature collapse. This supports the existence of interference under unified training, but it still does not directly quantify feature collapse at the representation level. As “feature collapse” is a central claim of this paper, more explicit feature-space analysis are needed.
> >
> > 2. The concern about the terminology of “Feature Space Reconstruction” remains. The method appears closer to constrained adaptation of an existing representation space than to reconstruction. I encourage the authors to revise this wording to avoid overclaiming.
> >
> > 3. As for semantic and artifact entanglement, my question comes from the paper’s own findings that strict orthogonality is suboptimal. I agree that completely addressing this issue may be beyond the scope of this paper, but some preliminary analysis or a more explicit discussion would strengthen methodological interpretation.

---

> > > ### Author Response · Authors · 2026-04-04
> > >
> > > Thank you for your feedback. We would like to clarify your three concerns in below.
> > >
> > > For Q1, we answer this in two perspectives.
> > >
> > > 1. As you previously acknowledged, we've confirmed the existence of collapse through qualitative experiments, which identifies why existing models fail. This evidence supports our logical progression from the heterogeneous artifact phenomenon to the SICA solution. Furthermore, our subsequent analysis in paper does not utilize the quantitative metrics. **Thus, the manuscript's logical integrity and completeness do not depend on quantitative experiments**.
> > > 2. Conducting quantitative experiments lacks a standardized protocol, as variables like base models, layers, and metrics (e.g., linear probing or SVD) significantly alter results, and specific quantitative numerical values do not change our paper's conclusions. We invite the reviewer to specify a preferred detailed setup. **Otherwise, a general quantitative requirement remains impractical to implement**.
> > >
> > > For Q2, "reconstruction" follows several widely accepted protocols. One such approach involves transforming from a non-collapsed space, as demonstrated in the following established works [1-3]. In our context, **SICA reconstructs CLIP's original semantic space into a discriminative artifact feature space**. To eliminate ambiguity, we will explicitly clarify this terminology in the revised manuscript.
> > >
> > > For Q3, we answer this in two perspectives.
> > >
> > > 1. Our observation that strict orthogonality is suboptimal was intended to demonstrate that SICA effectively reconstructs the artifact feature space from CLIP's semantic space by allowing flexibility, rather than enforcing rigid constraints. We believe there may be a misunderstanding regarding this point.
> > > 2. The entanglement **does not affect the correctness or logical self-consistency of our conclusions at all**. We invite the reviewer to specify which conclusion you believe is compromised by entanglement. Nevertheless, to improve readability, we will add a discussion on semantic-artifact relationships across different domains (e.g., [4]) in the revised related work section.
> > >
> > > [1] Unsupervised embedding adaptation via early-stage feature reconstruction for few-shot classification, ICML21
> > >
> > > [2] Anti-aliasing semantic reconstruction for few-shot semantic segmentation, CVPR21
> > >
> > > [3] Pre-training molecular graph representation with 3d geometry, ICLR22
> > >
> > > [4] Breaking semantic artifacts for generalized ai-generated image detection, NIPS24

---

### Official Review · Reviewer_FVbx · 2026-03-13

**Soundness:** 3
**Presentation:** 4
**Significance:** 2
**Originality:** 3
**Overall Recommendation:** 4
**Confidence:** 5

**Summary:**

The authors propose a monolithic model, SICA to address forgery detection in four kinds of modalities - Deepfake, AIGC, IMDL and Document forgeries. They also propose a combined dataset OpenMMSec with all four kinds of images to demonstrate SOTA performance.

**Compliance With Llm Reviewing Policy:**

Affirmed.

**Final Justification:**

After considering the rebuttal and follow-up discussion, my overall assessment remains largely unchanged.

The authors have strengthened the empirical evaluation, particularly by adding robustness experiments under multiple post-processing operations, which improves confidence in the model’s generalization. The clarification regarding the role of subspace updates and the distinction from standard fine-tuning is also helpful.

However, several key concerns remain unresolved. The construction of the OpenMMSec dataset still lacks sufficient detail regarding data integration, preprocessing, and potential biases, limiting reproducibility. Additionally, the methodological justification for handling varying forgery scales and the choice of similarity metrics remains somewhat qualitative. The attention map visualizations, while supplemented, are still not fully convincing in demonstrating meaningful alignment with manipulated regions.

Overall, I find the paper technically solid with promising ideas, but with limitations in reproducibility and interpretability that somewhat reduce its impact. I therefore maintain my original score of 4 (weak accept).

**Key Questions For Authors:**

- Why trying to combine forgery detection for the four modalities? Are there common artifacts in the generation of manipulation for each modality that the model is trying to capture? How does the model compare forgeries of different sizes in the different modalities.
- Can the authors use methods other than cosine similarity for direction updates? What is the rationale for choosing different cosine similarities in different subspaces?

**Limitations:**

Mentioned in Impact Statement

**Strengths And Weaknesses:**

Strengths:
- The authors tried to reconstruct a balance between unified and discriminative artifact for generalized forgery detection across four modalities. This is a hard task. The t-sne plots in Fig 1 seems promising.
- Spectral analysis of low-rank adaptation for artifact learning establishes geometric relationship between SICA and FFT is innovative.
- The experimental evaluation is very comprehensive with 98 forgery types and 15 SOTA baselines.
- The paper is very well written, organized, easy to understand and the figures have high quality illustrations to support.

Weaknesses:
- The OpenMMSec dataset generation details is not mentioned. What is the source of the images? How do the authors do image collection, mask generation, quality check in images? Are there any biases in the dataset? Cannot be reproduced with mentioned details in Section 3.
- No robustness analysis for the proposed model or proposed dataset reported.

---

> ### Author Rebuttal · Authors · 2026-03-30
>
> We deeply appreciate the reviewer's insightful comments and the encouraging recognition of our work. Please find our responses to your concerns below.
>
> - Concerns about the dataset.
>
> Thank you for the suggestion. We clarify in Line 181 that all images and masks in OpenMMSec are sourced from existing public forensic datasets across four domains, with a complete list provided in Appendix D.2. We select high-quality datasets within four domains and **ensure diverse faking types, balanced data volume, and rich image sources**, effectively mitigating bias. We will refine the construction details in the final version to improve readability.
>
> - "No robustness analysis for the proposed model or proposed dataset reported."
>
> We randomly sampled 2,500 real and 2,500 fake images in OpenMMSec to evaluate SICA's robustness against two SOTAs across four post-processing methods: Gaussian Blur, Gaussian Noise, JPEG Compression, and Resize. The results, provided in the anonymous link (https://anonymous.4open.science/r/some_pic-7037/robust.png), demonstrate SICA's superior robustness under varying post-processing conditions.
>
> - "Why trying to combine forgery detection for the four modalities?"
>
> First, the unified FID task was formally defined and proposed in [1], which explicitly established the setting of unified detection across these four domains. Second, other recent works [2] follow this configuration, representing a clear consensus in the current literature. Finally, beyond community conventions, we contend that these four subdomains are the most widely studied in the fake image field, and successfully unifying them demonstrates a scalable paradigm that can theoretically integrate any future forensics tasks using the same methodology.
>
> [1] Forensichub: A unified benchmark & codebase for all-domain fake image detection and localization, NIPS25
>
> [2] UniShield: An Adaptive Multi-Agent Framework for Unified Forgery Image Detection and Localization
>
> - "Are there common artifacts in the generation of manipulation for each modality that the model is trying to capture?"
>
> Are you inquiring whether common artifacts exist across the four domains? Our perspective is as follows. We acknowledge the existence of common artifacts across domains, as learning directly on unified data yields a baseline level of performance. However, the heterogeneous phenomenon of artifacts causes the shared feature space to collapse, limiting overall accuracy. If this does not accurately reflect your question, please let us know and we will provide further clarification.
>
> - "How does the model compare forgeries of different sizes in the different modalities."
>
> Are you inquiring about how the model handles varying forgery sizes across different domains? Our perspective is as follows. We attribute this to SICA's ability to reconstruct the artifact feature space via semantic induction. For instance, AIGC involves full-image generation where forgeries span the entire image, whereas Doc tampered text regions are typically much smaller. As shown in Fig. 1 (right) in the paper, SICA's reconstructed space exhibits clearer separation and more distinct decision boundaries between AIGC and Doc compared to the original CLIP space. This demonstrates that the success in detecting forgeries of different sizes stems from a superior, reconstructed feature space. If this does not accurately reflect your question, please let us know and we will provide further clarification.
>
> - "Can the authors use methods other than cosine similarity for direction updates?"
>
> We also evaluate Grassmannian Distance by comparing the principal angles between the SVD-decomposed subspaces of original and updated weights. Mathematically, this captures subspace alignment complementary to cosine similarity. The results we provide in anonymous link (https://anonymous.4open.science/r/some_pic-7037/grassmannian.png) confirm that SICA’s updates are geometrically distant from semantic directions with larger distance, consistent with our cosine similarity findings and further validating the reconstruction of a discriminative artifact space.
>
> - "What is the rationale for choosing different cosine similarities in different subspaces?"
>
> To clarify, are you inquiring about the rationale in both the left and right subspaces? Our perspective is as follows. The rationale for analyzing both subspaces is that they represent distinct functional aspects of the model's weight adaptation. The left subspace reflects the distribution of output representations, where lower similarity indicates the reconstruction of a unified-yet-discriminative feature space. The right subspace reflects the model's dependence on input features, where lower similarity here confirms that SICA effectively circumvents semantic shortcuts by shifting focus toward artifact-related patterns. If this does not accurately reflect your question, please let us know and we will provide further clarification.

---

> > ### Author Rebuttal · Reviewer_FVbx · 2026-04-04
> >
> > Thank you for the rebuttal. The rebuttal improves the paper in several important aspects. The added robustness experiments under multiple post-processing operations and evaluation on semantically similar FLUX-generated images strengthen the empirical claims and demonstrate improved generalization. The clarification regarding the role of singular vectors and the distinction between SICA and standard fine-tuning is also helpful.
> >
> > However, several concerns remain:
> > - The paper still lacks essential details about the construction of the OpenMMSec dataset (data sources, annotation pipeline, and potential biases), which limits reproducibility. Additionally, key methodological questions—such as the choice of cosine similarity and behavior across different forgery scales—were not addressed.
> > - Importantly, the attention map visualizations are not fully convincing: ground-truth masks are not overlaid, making it difficult to assess localization quality, and the highlighted regions appear diffuse and not clearly aligned with manipulated areas. This weakens the interpretability claims.
> > - Overall, while the rebuttal strengthens empirical validation, gaps in reproducibility, methodological justification, and qualitative evidence remain. I encourage the authors to address these issues to be considered for acceptance.

---

> > > ### Author Response · Authors · 2026-04-04
> > >
> > > Thank you for your valuable feedback and constructive suggestions, which help improve the clarity and quality of our work. We especially appreciate your rigorous examination of our dataset construction and technical motivations, which has helped us further clarify the contributions of our work. We would like to address your concerns in below.
> > >
> > > **Q1**: Concern about the construction of the OpenMMSec dataset.
> > >
> > > - **data sources, annotation pipeline:** As previously stated in our rebuttal, we clarify in Line 181 in paper that all images and masks in OpenMMSec are sourced from existing public forensic datasets across four domains (e.g. FaceForensics++, GenImage), with a complete list provided in Appendix D.2. The data sources and annotation pipelines are detailed in their respective original papers. To avoid redundancy and respect the contributions of the original authors, we have omitted these descriptions from our manuscript. Additionally, the original dataset source for each image is clearly labeled in the JSON file provided with OpenMMSec.
> > >
> > > - **potential biases:** As previously stated in our rebuttal, we select high-quality datasets within four domains and **ensure diverse faking types, balanced data volume, and rich image sources**, effectively mitigating bias. Despite efforts to minimize bias, potential bias persists. As addressed in our rebuttal to Reviewer TRbs, the real-fake disparity in Doc datasets stems from strict privacy and copyright constraints on real images. **Importantly, such imbalances are prevalent in established forensic benchmarks (e.g., [1,2]) and do not compromise evaluation validity**.
> > >
> > > We will refine the construction details in the final version to improve readability.
> > >
> > > [1] Faceforensics++: Learning to detect manipulated facial images, ICCV19
> > >
> > > [2] DF40: Toward Next-Generation Deepfake Detection, NIPS24
> > >
> > > **Q2:**  The choice of cosine similarity and behavior across different forgery scales.
> > >
> > > We addressed these points in our previous rebuttal (please see "How does the model compare forgeries of different sizes in the different modalities.", "Can the authors use methods other than cosine similarity for direction updates?" and "What is the rationale for choosing different cosine similarities in different subspaces?").
> > >
> > > For instance, we utilize cosine similarity as the standard metric for directional alignment and add Grassmannian Distance complementary to cosine similarity provide in anonymous link (https://anonymous.4open.science/r/some_pic-7037/grassmannian.png). Similarly, "forgery scales" is not a standardized term and allows for multiple interpretations, such as image resolution or the dimensions of manipulated regions. **Please clarify your specific meaning so we can conduct further experiments and provide explanations to address your concerns**.
> > >
> > > **Q3:** Concern about the attention map visualizations.
> > >
> > > **First of all, We clarify that our target task is image-level detection rather than pixel-level localization.** Established Deepfake and AIGC benchmarks typically lack pixel-level masks, as Deepfake manipulations target faces and AIGC images are entirely generated.
> > >
> > > The purpose we retained masks from IMDL and Doc is for the future localization research (described in Line 200 of paper). Nevertheless, we agree masks help with interpretability and have added the visualized SICA’s attention maps against these masks in the anonymous link (https://anonymous.4open.science/r/some_pic-7037/attention_v2.png). Although **detection models naturally yield less precise between attention maps and masks than localization models**, these maps demonstrate that **SICA identifies subdomain-specific artifacts consistent with prior studies** (see response to Reviewer TRbs). This confirms **SICA successfully reconstructs the artifact feature space by accommodating heterogeneous artifacts without cross-domain interference**.

---

### Official Review · Reviewer_13Rf · 2026-03-15

**Soundness:** 3
**Presentation:** 3
**Significance:** 2
**Originality:** 2
**Overall Recommendation:** 4
**Confidence:** 4

**Summary:**

The authors consider fake image detection across four image forensic subdomains. Typically, approaches rely on an ensemble of models and having a single approach does not perform as well on these tasks. The authors hypothesize that different types of fake images have different kinds of artifacts which they term as the "heterogeneous phenomenon". Due to the heterogenity, the models pick up on simpler, common artifacts thereby learning inferior features. The authors introduce a soft-constrained adaptation technique called Semantic-Induced Constrained Adaptation (SICA) to serve as a monolith fake image detector. Since artifacts are heterogeneous, the authors work with the hypothesis that the presence of some semantic information would guide the detector to pick up on these different types of artifacts. The methods efficacy is demonstrated through results on OpenMMSec dataset. They also perform a spectral analysis to show that the weight updates do not have a great overlap with the principal subspace.

**Compliance With Llm Reviewing Policy:**

Affirmed.

**Final Justification:**

The papers cited by the authors confirms that the semantic interpretation of singular vectors with the highest singular value is a well established and trustworthy result. The analysis on parameter count is helpful in clarifying that the method does more than just controlling parameter count. Furthermore, the reconstruction based results are also promising and shows that the method can separate reconstructions from their real images showing that the method indeed picks up on lower-level semantics and that the semantic prior just helps with the model learning the same.

The paper has good experimental rigor and an interesting perspective warranting acceptance.

**Key Questions For Authors:**

Given the emphasis on some "semantic" features, I am curious to see how this method would fare when trying to separate real images from their reconstructed one (use FLUX and perform DDIM)

**Limitations:**

Yes

**Strengths And Weaknesses:**

**Soundness**
Strengths: Method works well, outperforms a large number of baselines on the OpenMMSec subspace. Experiments demonstrate clear advantages over other CLIP-finetuning techniques such as FFT and Effort.

Weaknesses: Line 66 claim is not backed up by Fig 1, to me that looks like a significant overlap between the AIGC and IMDL distributions. More fundamentally, the main method is motivated as providing a soft-prior by using a low-rank adaptation for training the detector. However, it is not clear whether the "semantic" prior helps. Interpreting the first few singular directions as "semantic directions" does not seem completely correct to me. I am curious if this is because of a semantic prior induced by LoRA training or just having the correct parameter count in a less constrained manner (compared to Effort), this could be validated by performing finetuning without LoRA but using same number of parameters as the LoRA version (freeze every other layer etc). Nitpick: Line 151-155, it is preferrable to not list strong validation as a contribution, I feel it is required for any paper.

**Presentation**
Strength:The paper is well written.


**Significance**
Strengths: The method outperforms a lot of common baselines and would be a useful method to benchmark against for future techniques.

Weaknesses: There are no results analyzing the sensitivity to post-processing. This is extremely important for a method to be practically useful.

**Originality**

Since the core motivation relies on somewhat nebulous concepts such as preserving semantic features and assumes that the first
k singular vectors encode this information, the central claim is difficult to verify empirically. Without clearer evidence that these singular vectors indeed correspond to semantic information, it becomes hard to determine whether the observed improvements stem from the proposed mechanism or simply from the regularization effect of reducing the number of trainable parameters, similar to applying LoRA to control overfitting..

---

> ### Author Rebuttal · Authors · 2026-03-30
>
> We sincerely thank the reviewer for the time and constructive feedback, which have greatly helped us improve the manuscript.
>
> **We believe your primary concern actually stem from the misunderstanding of interpretation of singular vectors**. However, our characterization of the first few singular vectors as "semantic information" follows a well-established interpretive practice in the literature:
>
> 1. **Prior works show that principal/singular vectors in CLIP- or ViT-related spaces are often aligned with semantically meaningful variations.** This is supported by: [1] demonstrates that features represented by singular vectors are interpretable and semantic, while [2] shows that top principal components in CLIP capture consistent abstract knowledge across models.
> 2. In Effort, the model learns fake patterns (artifacts) in subspaces orthogonal to the CLIP principal directions, where artifacts in forensics are typically considered non-semantics[3].
>
> Therefore, our claim of singular vectors as semantics grounds on solid previous findings and can support our motivation of SICA reconstructing the artifact feature space based on the CLIP semantic manifold. The fact that singular vectors encode semantics has been well-established in study of vision foundation and generative models, yet remains underexplored within the forensic community.
>
> **Likewise, your other concerns also stems from the misunderstanding that we use LoRA merely to control overfitting.** On one hand, SICA acts as a structural reconstruction paradigm that evolves a well-established semantic manifold into a discriminative artifact space, rather than merely providing a soft prior for learning from scratch. On the other hand, we supplement two experiments: Linear Probing (freezing CLIP weights, training the classification head) and Partial (training the last attention layer in CLIP-ViT with parameters comparable to LoRA). The table below compares their parameters and average AUC. Results show that both Linear Probing (fewer parameters) and Partial (similar parameters) significantly underperform compared to SICA. This gap underscores SICA's effectiveness in successfully reconstructing the artifact feature space. **Therefore, SICA is totally irrelevant to tuning the overfitting of CLIP on FID.**
>
> |Model|SICA (Ours)|Linear Probing|Partial|Effort|Fully Finetune|
> |:-:|:-:|:-:|:-:|:-:|:-:|
> |Parameters|1.18M|768|1.05M|0.2M|427M|
> |AUC|0.875|0.772|0.807|0.842|0.801|
>
> Please find our responses to the remaining concerns below.
>
> - "Line 66 claim is not backed up by Fig 1, to me that looks like a significant overlap between the AIGC and IMDL distributions."
>
> The overlap occurs because both AIGC and IMDL target within natural scenes. Their semantic gap is smaller than that between Deepfake and Doc. We will refine the description in the final version to more accurately.
>
> - "it is preferrable to not list strong validation as a contribution"
>
> Thank you for the suggestion, and we will revise this contribution in the final version.
>
> - "There are no results analyzing the sensitivity to post-processing."
>
> We randomly sampled 2,500 real and 2,500 fake images in OpenMMSec to evaluate SICA's robustness against two SOTAs across four post-processing methods: Gaussian Blur, Gaussian Noise, JPEG Compression, and Resize. The results, provided in the anonymous link (https://anonymous.4open.science/r/some_pic-7037/robust.png), demonstrate SICA's superior robustness under varying post-processing conditions.
>
> - "how this method would fare when trying to separate real images from their reconstructed one"
>
> We randomly selected 5 images per class from ImageNet-1K and generated semantically similar fake counterparts using FLUX with DDIM. This 10K image test set (examples in the anonymous link https://anonymous.4open.science/r/some_pic-7037/flux.png) was used to evaluate SICA against two SOTA methods. Results below show that SICA maintains superior detection performance even on semantically similar images, demonstrating the strong generalization of our reconstructed feature space.
>
> |Model|SICA (Ours)|Effort|Trufor|
> |:-:|:-:|:-:|:-:|
> |AUC|0.8215|0.7971|0.7687|
>
> [1] Dissecting query-key interaction in vision transformers, NIPS24
>
> [2] Text-to-concept (and back) via cross-model alignment, ICML23
>
> [3] Forensichub: A unified benchmark & codebase for all-domain fake image detection and localization, NIPS25

---

> > ### Author Rebuttal · Reviewer_13Rf · 2026-04-03
> >
> > I thank the authors for taking the time to address my concerns. The papers cited by the authors confirms that the semantic interpretation of singular vectors with the highest singular value is a well established and trustworthy result. I encourage the authors to include the citation in the final version of the paper.
> >
> > I do want to clarify that I never interpreted the method as using LoRA to control underfitting, but the current analysis is further helpful in further clarifying the same. Furthermore, the reconstruction based results are also promising and shows that the method can separate reconstructions from their real images showing that the method indeed picks up on lower-level semantics and that the semantic prior just helps with the model learning the same.
> >
> > Therefore, I will increase my score by 1 and recommend acceptance of this paper.

---

> > > ### Author Response · Authors · 2026-04-07
> > >
> > > Thank you for reconsidering our work and upgrading the score. We sincerely appreciate your constructive feedback throughout the discussion, which has been instrumental in refining the technical clarity and logical rigor of our manuscript. We will fully incorporate your suggestions into the final version.

---

### Decision · Program_Chairs · 2026-04-30

**Decision:**

Accept (regular)

**Comment:**

This paper finally received one Weak Reject, two Weak Accepts, and one Accept. After the rebuttal, Reviewer 13Rf marked concerns as fully resolved and raised the score by one, moving from Weak Accept to a clear recommendation for acceptance. Reviewer TRbs also marked concerns as fully resolved and maintained Accept. Reviewer FVbx remained partially resolved, citing remaining gaps in dataset reproducibility, methodological justification, and qualitative interpretability. Reviewer QUXG remained partially resolved as well: the score was raised (notably reflecting improved appreciation of the contribution beyond a naive “CLIP + LoRA” reading and added robustness evidence), but the reviewer still requested more direct representation-level support for feature collapse and tighter wording around “feature space reconstruction.”

Considering the net positive shift across reviewers, the strong empirical additions in the rebuttal (robustness controls, reconstruction-style evaluation, and clarifying analyses), and the fact that the remaining objections are largely addressable through revision (terminology calibration, reproducibility documentation, and clearer mechanistic framing) rather than fundamental invalidity, the AC recommends accepting this paper. Congratulations!

When preparing the final version, the authors should incorporate the rebuttal clarifications into the main text, complete OpenMMSec construction and split details, explicitly discuss what evidence is direct vs. indirect, and task-appropriate framing of attention-map interpretability for image-level detection.